



# MAX-DOAS measurements of NO₂, SO₂, HCHO and BrO at the Mt. Waliguan WMO/GAW global baseline station in the Tibetan Plateau

Jianzhong Ma[1], Steffen Dörner[2], Sebastian Donner[2], Junli Jin[3], Siyang Cheng[1], Junrang Guo[1], Zhanfeng Zhang[4], Jianqiong Wang[4], Peng Liu[4], Guoqing Zhang[4], Janis Pukite[2], Johannes Lampel[2,5], Thomas Wagner[2]

[1]State Key Laboratory of Severe Weather & CMA Key Laboratory of Atmospheric Chemistry, Chinese Academy of Meteorological Sciences, Beijing, China

[2]Max Planck Institute for Chemistry, Mainz, Germany

[3]CMA Meteorological Observation Centre, Beijing, China

[4]Waliguan Observatory, Qinghai Meteorological Bureau, Xining, China

[5]now at: Airyx GmbH, Eppelheim, Germany

*Correspondence to*: Thomas Wagner (thomas.wagner@mpic.de) and Jianzhong Ma (majz@cma.gov.cn)

**Abstract.**

Mt. Waliguan Observatory (WLG) is a World Meteorological Organization (WMO)/Global Atmosphere Watch (GAW) global baseline station in China. WLG is located at the northeastern part of the Tibetan plateau (36°17′N, 100°54′E, 3816 m a.s.l.) and has a representativeness of the pristine atmosphere over the Eurasian continent. We made long-term ground-based MAX-DOAS measurements at WLG during the years 2012–2015. In this study, we retrieve the differential slant column densities (dSCDs) and estimate the tropospheric background mixing ratios of different trace gases, including NO₂, SO₂, HCHO and BrO, using the measured spectra at WLG. We find averaging of 10 original spectra to be an 'optimum option' for reducing both the statistical error of the spectral retrieval and systematic errors in the analysis. We retrieve the dSCDs of NO₂, SO₂, HCHO and BrO from measured spectra at different elevation angles under clear sky and low aerosol load conditions at WLG. By performing radiative transfer simulations with the model TRACY-2, we establish approximate relationships between the trace gas dSCDs at 1° elevation angle and the corresponding average tropospheric background volume mixing ratios. Mixing ratios of these trace gases in the lower troposphere over WLG are estimated to be between about 5 ppt (winter) and 70 ppt (summer) for NO₂, fall below 0.5 ppb for SO₂, range between about 0.3 and 0.7 ppb for HCHO, and be close to ~0 ppt for BrO. Our study provides valuable information and data set for further investigating tropospheric background levels of these trace gases and their relationship to anthropogenic activities.

## 1 Introduction

Nitrogen oxides (NOₓ ≡ NO + NO₂), sulfur dioxide (SO₂) and formaldehyde (HCHO) are important traces gases in the troposphere. Both NOₓ and HCHO participate in the control of the strong oxidant O₃, which is an indicator of photochemical smog, and the strongest atmospheric oxidizing agent OH, which determines the lifetimes of many gaseous pollutants and greenhouse gases in the atmosphere (Seinfeld and Pandis, 2006;Ma et al., 2012;Lelieveld et al., 2016). NOₓ and SO₂ are gaseous precursors of nitrate and sulfate aerosols, and large amounts of these aerosols can result in haze pollution and exert a strong negative radiative forcing on the climate change (Seinfeld and Pandis, 2006;Forster et al., 2007;Ma et al., 2010). NOₓ and SO₂ are released from various anthropogenic emission sources, e.g., the burning of coal, oil, gas, wood, and straw (Granier et al., 2011;Zhao et al., 2012). NOₓ is also emitted via natural processes including lightning and microbial activities in soils (Lee et al., 1997). Nitric oxide (NO) dominates NOₓ released from these sources, but it can be quickly converted to nitrogen dioxide (NO₂) by reaction with ozone (O₃) in the atmosphere. Natural sources of SO₂ in the troposphere include volcanic eruptions and the atmospheric oxidation of dimethyl sulfide (DMS: CH₃SCH₃) emitted from the ocean (Dentener et al., 2006). HCHO in the remote atmosphere is produced through the oxidation of methane (CH₄) and non-methane volatile





compounds (NMVOCs), and it is also emitted from anthropogenic combustion processes, biomass burning and natural vegetation (Stavrakou et al., 2009).

Global emissions and atmospheric abundance of gaseous pollutants (e.g., $NO_x$ and $SO_2$) and aerosols have changed significantly over the past few decades as revealed predominantly by satellite observations (e.g., De Smedt et al., 2015;Xing et al., 2015;Bauwens et al., 2016;Fioletov et al., 2016;Krotkov et al., 2016;Klimont et al., 2017;Li et al., 2017;Georgoulias et al., 2018;Hammer et al., 2018;Ziemke et al., 2018). Worldwide ground-based monitoring the concentrations and trends of trace gases and aerosols in the atmosphere is essential for the validation of and filling gaps in satellite observations, in order to quantitatively assess the impacts of atmospheric composition change on global air quality and climate changes (Stohl et al., 2015;De Mazière et al., 2018). The international global measurement networks have been set up sequentially over the past decades to establish long-term databases for detecting changes and trends in the chemical and physical state of the atmosphere. Among the networks are the Global Atmosphere Watch (GAW) program of the World Meteorological Organization (WMO) and the Network for the Detection of Atmospheric Composition Change (NDACC), and in the latter measurements are performed mainly by ground-based remote-sensing techniques (De Mazière et al., 2018). In contrast to Europe and North America, the stations under the networks in Asia, especially in the remote areas, are very sparse.

The China Global Atmosphere Watch Baseline Observatory at Mt. Waliguan (WLG) is an in-land GAW baseline station affiliated to WMO. The site (3816 m a.s.l.) is located at the northeastern part of the Tibetan plateau, being representative of the pristine atmosphere over the Eurasian continent. Air masses at WLG are highly representative of the remote free troposphere (Ma et al., 2002a). Long-term measurement results have shown a summer peak in the seasonal cycle of surface ozone at WLG, and the causes have been a subject of discussion in several studies (Zhu et al., 2004;Ma et al., 2005;Ding and Wang, 2006;Li et al., 2009). Previous model simulations constrained by measured mixing ratios of ozone and its precursors (including $NO_2$) indicated a net destruction of ozone during summertime at WLG (Ma et al., 2002a). In contrast, new insights from model calculations based on more recent measurements (including NO) showed that ozone is net produced by in situ photochemistry at WLG during both late spring and summer (Xue et al., 2013). The difference in the sign of ozone production between the two studies is mainly due to the fact that higher $NO_x$ levels were measured in the year 2003 using a chemiluminescence analyzer (Xue et al., 2013) than in the year 1996 by the filter sampling and ion chromatography analysis (Ma et al., 2002a). It is not determined if the different levels of $NO_x$ were caused by the uncertainties in the measurements, especially from the filter sampling, or there was practically an increasing trend of $NO_x$ at WLG from 1996 to 2003. Similar to the earliest measurement of $NO_2$, $SO_2$, HCHO and other reactive gases were also measured by filter or canister sampling method at WLG at irregular time (Mu et al., 2007;Meng et al., 2010;Lin et al., 2013). It is interesting and necessary to start a new measurement program with advanced technique at WLG for the purpose of precisely monitoring the levels and trends of atmospheric composition in the global pristine atmosphere.

The multi-axis differential optical absorption spectroscopy (MAX-DOAS) has the potential to retrieve the vertical distributions of trace gases and aerosols in the immediate vicinity of the station from the scattered sunlight measured at multiple elevation angles (Hönninger and Platt, 2002;Bobrowski et al., 2003;Van Roozendael et al., 2003;Hönninger et al., 2004;Wittrock et al., 2004). As relatively simple and cheap ground-based instrumentation, the UV-visible MAX-DOAS will be included in the certified NDACC measurement technique for the observation of lower-tropospheric $NO_2$, HCHO and $O_3$ (De Mazière et al., 2018). Successful measurement and retrieval of trace gases (e.g., $NO_2$, $SO_2$ and HCHO) depend on various factors, including their molecular absorption features and atmospheric abundances as well as the atmospheric visibility and instrumental signal/noise ratio. In contrast to extensive ground-based measurements of $NO_2$, $SO_2$ and HCHO in rural and urban areas worldwide, including highly polluted areas in eastern China (e.g., Ma et al., 2013;Hendrick et al., 2014;Wang et al., 2014;Jin et al., 2016;Wang et al., 2017), measurements of these trace gases by MAX-DOAS in the remote areas have been very sparse (Wittrock et al., 2004;MacDonald et al., 2012;Schreier et al., 2016). The $NO_2$ mixing ratio under the clean background conditions in the boundary layer of the Artic was estimated to be 30 pptv (Wittrock et al., 2004). HCHO was observed at a maximum concentration of 4.5 ppbv in the tropical forest (MacDonald et al., 2012) and at an average mixing ratio of 7 ppbv in the rural area of Southern China (Li et al., 2013). The monthly mean mixing ratios of free tropospheric $NO_2$ (HCHO) under clear-sky conditions were measured in the range of 60–100 pptv (500–950 pptv) at the mid-latitude site Zugspitze (2650 m a.s.l.) in Germany and 8.5–15.5 pptv (255–385 pptv) at the tropical site Pico Espejo





(4765 m a.s.l.) in Venezuela (Schreier et al., 2016). The MAX-DOAS technique has been applied to monitor the absolute column densities and plumes of $SO_2$ from large volcano eruptions (e.g., Lübcke et al., 2016;Tulet et al., 2017), but measuring $SO_2$ in the background free troposphere still remains challenging.

Bromine oxide (BrO) can play an import role in the catalytic destruction of ozone in the remote troposphere (Platt and Hönninger, 2003;von Glasow and Crutzen, 2007). The earliest MAX-DOAS measurements were focused on the retrieval of the mixing ratio levels and vertical profiles of BrO in the boundary layer of the Artic, salt lake and marine areas (Hönninger and Platt, 2002;Stutz et al., 2002;Leser et al., 2003;Frieß et al., 2004;Saiz-Lopez et al., 2004). Measurement results showed that the BrO mixing ratio could reach up to 30 pptv in the Actic and 10 pptv in the marine boundary layer (Platt and Hönninger, 2003;Martin et al., 2009). BrO in the free troposphere at the global scale was estimated to be at a level of 0.5–2 pptv based on space-borne, ground-based and sounding measurements by DOAS technique (Harder et al., 1998;Fitzenberger et al., 2000;Richter et al., 2002;Van Roozendael et al., 2002;Hendrick et al., 2007;Theys et al., 2007;Werner et al., 2017). Model calculations indicated that inorganic bromine can influence the chemical budgets of ozone in the free troposphere to a considerable extent, reducing $O_3$ concentration locally by up to 40% (von Glasow et al., 2004;Lary, 2005;Yang et al., 2005;Yang et al., 2010). Until now, measurements of BrO and related species by ground-based MAX-DOAS have been frequently carried out in the Arctic (Peterson et al., 2015;Peterson et al., 2017;Simpson et al., 2017;Luo et al., 2018), Antarctic (Wagner et al., 2007a;Roscoe et al., 2012;Prados-Roman et al., 2018) and coastal atmosphere (Coburn et al., 2011). To our knowledge, no MAX-DOAS measurements have been reported for BrO on continents other than polar, salt lake and coastal areas.

We made long-term ground-based MAX-DOAS measurements at WLG during the years 2012–2015. For this study we analyzed the measured spectra to retrieve estimates or at least upper limits of the free tropospheric background mixing ratios of different trace gases, including $NO_2$, $SO_2$, HCHO and BrO, from MAX-DOAS measurements at WLG. Large effort was spent on the spectral analysis, because in spite of the rather long atmospheric light paths at high altitude the respective trace gas absorptions are close to or below the detection limit. In Sect. 2, we give a description of the WLG measurement site and the MAX-DOAS instrument used in the study. In Sec. 3, we analyze the meteorological conditions and their seasonal variations over the WLG site. Sect. 4 describes the method and settings we used in the spectral retrieval. Sect. 5 introduces the radiative transfer simulations we performed for in-depth analysis of measurement data. Sect. 6 describes the methods to filter the measurement data for the clear sky and low aerosol load conditions. In Sect. 7, we provide the differential slant column density values of the investigated trace gases and their corresponding tropospheric background mixing ratios over WLG. Conclusions are given in Sect. 8.

## 2 Description of the measurement site and the instrument

### 2.1 WLG station

The WLG station is sited at the top of Mt. Waliguan (36°17′N, 100°54′E, 3816 m a.s.l.), located in Qinghai Province of China (Fig. 1a). It is one of the WMO/GAW global baseline stations and only one in the hinterland of the Eurasian continent. Mt. Waliguan is an isolated mountain with an elevation of about 600 m relative to the surrounding landmass, being surrounded by highland steppes, tundra, deserts, and salt lakes (Fig. 1b). With a low population density of about 6 capitals km$^{-2}$ and hardly any industry within 30 km, WLG has the advantage to be rather isolated from industry, forest and population centers. It is relatively dry, windy and short of precipitation with a typical continental plateau climate (Tang et al., 1995). Xining City (the capital of Qinghai Province, located about 90 km northeast of WLG) and Lanzhou City (the capital of Gansu Province, located about 260 km away to its east) are considered as the nearest large pollution sources that may have impacts on the WLG site (Wang et al., 2006). There are several high mountains (~4000 m a.s.l.) between Xining and Mt. Waliguan. Total column ozone, surface ozone, solar radiation, precipitation chemistry, greenhouse gases, aerosol optical depth, and aerosol scattering/absorption coefficient together with basic meteorological parameters have become operational measurement items at WLG in succession since the year 1991 (Tang et al., 1995).

In addition to routine observations, intensive measurements and model analyses were performed to investigate the regional/global representativeness of WLG and the effects of chemical transformations, physical and transport processes





involved for various atmospheric composition, e.g., surface ozone (Ma et al., 2002a;Zhu et al., 2004;Ma et al., 2005;Ding and Wang, 2006;Wang et al., 2006;Li et al., 2009;Xue et al., 2013;Xu et al., 2016;Xu et al., 2018), short-lifetime reactive gases (such as NO, $NO_2$, $SO_2$, CO, $H_2O_2$, $HNO_3$, HCHO, other carbonyls, and non-methane hydrocarbons (NMHCs)) (Ma et al., 2002a;Wang et al., 2006;Mu et al., 2007;Meng et al., 2010;Xue et al., 2011;Zhang et al., 2011;Lin et al., 2013;Xue et al.,

5    2013), greenhouse gases (Zhou et al., 2004;Zhou et al., 2005;Zhou et al., 2006;Zhang et al., 2008;Fang et al., 2013;Zhang et al., 2013;Fang et al., 2014;Liu et al., 2014;Zhang et al., 2015;Cheng et al., 2017;Cheng et al., 2018), persistent organic pollutants (Cheng et al., 2007), metal and isotopes (Lee et al., 2004;Zhou et al., 2005;Zhou et al., 2006;Lee et al., 2007;Wang et al., 2007;Liang et al., 2008;Fu et al., 2012;Liu et al., 2014;Zheng et al., 2015), and aerosols (Gao and Anderson, 2001;Ma et al., 2003;Kivekäs et al., 2009;Che et al., 2011;Zheng et al., 2015).

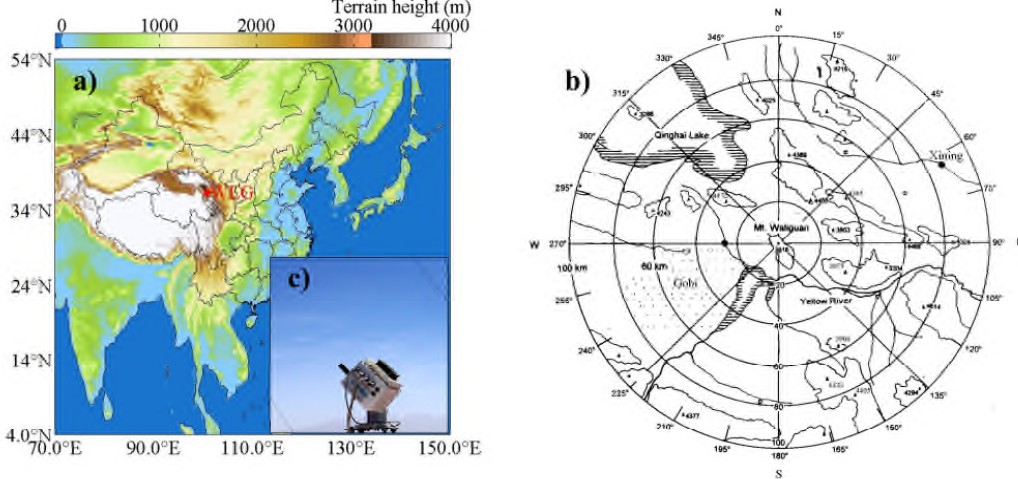

**Figure 1 (a)** Position of Mt Waliguan in East Asia. **(b)** Surrounding topography within 100 km distance from WLG. **(c)** The MAX-DOAS instrument installed at WLG.

**2.2 MAX-DOAS instrument**

We started the ground-based MAX-DOAS measurement program at WLG on 26 September 2010. An automated and compact (13 cm × 19 cm × 14 cm) Mini MAX-DOAS instrument from Hoffmann Messtechnik GmbH in Germany, which had ever been used at the Gucheng site in the North China Plain (Jin et al., 2016), was moved to and installed at WLG (Fig. 1c). This instrument is designed for the spectral analysis of scattered sunlight and the application of the MAX-DOAS

technique (Hönninger et al., 2004). The same type of instrument was used in previous studies, e.g. in Beijing and the surrounding area (Ma et al., 2013;Jin et al., 2016). The entrance optics, fiber coupled spectrograph and controlling electronics are hermetically sealed in a metal box of about 3 liter volume. A stepper motor, mounted outside the box, can rotate the whole instrument to control the elevation viewing angle, i.e., the angle between the horizontal and the viewing direction, and thus it can scan vertically at different elevation angles. The spectrograph covers a wavelength range of

290-437 nm and its entrance slit has a width of 50 μm. A Sony ILX511 charged coupled device (CCD) detects the light in 2048 individual pixels. The whole spectrograph is cooled by a Peltier stage to maintain a stable temperature of the optical setup and to guarantee a small dark current signal. The measurement process and spectra data logging are controlled by a laptop using the MiniMAX software package developed by Dr. Udo Frieß at the Institute of Environmental Physics, the University of Heidelberg in Germany.





The instrument was mounted on a bracket, fixed at the building roof, at an azimuth viewing direction exactly towards the north. After a winter of pilot run, we added heating elements to the outside of the instrument so that the temperature of the spectrograph could be kept at a stable but not very low value, e.g., -10 °C in winter. In other seasons, the temperature of the optical setup was set at a higher value, typically of 0 °C, below the ambient temperature. Dark current spectra were measured using 10000 msec and 1 scan, and electronic offset spectra with 3 msec and 1000 scans. Measurements of these signal spectra were made generally month by month or whenever the working temperature of the instrument was changed. Over the pilot run period in the years 2010 and 2011, measurements had been made with the same sequence of elevation angles as used at the Gucheng site in the North China Plain (Jin et al., 2016), with no elevation angles lower than 3° available. After the beginning of the year 2012, the elevation angles were set to be -1°, 0°, 1°, 2°, 3°, 5°, 10°, 20°, 30°, and 90° in a sequence. Unfortunately, during the data analysis it turned out that the elevation calibration was wrong by -4° (see Fig. A1 in the appendix). Thus finally only a few elevation angles from the original selection were found to be above the horizon. After the correction by -4° the remaining elevation angles are: 1°, 6°, 16°, 26°, and 86°. The exposure time for each elevation angle was about one minute. The data from three years of measurements over the period April 2012 through April 2015 are used for this study.

## 3 Meteorological conditions

We used the European Centre for Medium-Range Weather Forecasts (ECMWF) re-analysis data to investigate meteorological conditions over the WLG site, using data for 3-4 km altitude to represent the ground level at WLG. Figures 2, A2 and A3 present the daily mean values of selected meteorological variables over the years 2012-2015 and their climatically monthly mean values for the period over WLG. As shown in Fig. 2, the seasonal cycle is strong for these meteorological quantities. Specifically, temperature is high in summer (around 283K) and low in winter (around 265K), pressure is high in summer (around 643 hPa) and low in winter (around 635 hPa), wind speed is low in summer (around 3m/s) and high in winter (around 5m/s), and wind direction (0 means the wind is blowing from the North) is from south-east in summer (around 140°) and from west in winter (around 260°). These seasonal variation characteristics are the similar to those in earlier years at the station as reported in previous work (Tang et al., 1995). We evaluated the ECMWF reanalysis data for 3-4 km altitude using meteorological data from in situ measurements at the WLG station, and found that the ECMWF data are in good (temperature, pressure) and reasonable (wind speed and direction) agreement with in situ data (see Fig. A4).

We calculated the correlation coefficients among various meteorological variables for both monthly- and daily- averaged values (see Fig. A5). In addition to the height level of the station, we also investigated the ECMWF wind data for other altitudes, which might be important for the study of long range transport. There is a low correlation between wind directions below 5 km and above 6 km. Below 5 km the wind direction changes systematically between summer and winter. Above 6 km it is almost constant (around 270°) throughout the year with only a few sporadic exceptions. For monthly means (Fig. A5 left), high correlation is found between temperature and other variables, including pressure, tropopause height, wind speed above 5 km, and wind direction below 5 km (here it should be noted that the wind direction is usually in the range between 90 and 300°. Thus the derived correlations are not affected by the potential ambiguity of wind directions around 0°/360°). For daily means (Fig. A5 right), much lower correlation is found, but with the same general dependencies as for the monthly means. This indicates that the correlation is mainly determined by the seasonal cycle. However, still high correlation ($r^2 > 0.5$) is found for some daily mean quantities, such as temperature and tropopause height ($r^2=0.54$, $r=0.74$), wind speeds above 6km ($r^2>0.66$, $r> 0.81$), neighbouring wind speeds at all altitudes ($r^2>0.76$, $r>0.87$), and wind directions above 7 km ($r^2>0.69$, $r>0.83$).



**Figure 2 Left:** Time series of daily averaged values of different meteorological quantities at WLG station derived from ECMWF data (2012 to 2015). **Right:** The corresponding seasonal cycles. Time series of wind speed and wind directions at different altitude levels are provided in the appendix.



## 4 Spectral retrieval

In this section the settings of the spectral retrievals are described, and exemplary fit results are presented. At the WLG site usually the atmospheric trace gas absorptions are rather low. Thus the settings of the spectral analysis were optimised for low detection limits. In general this can be achieved by

    a) co-adding of individual spectra,

    b) using rather broad spectral ranges,

    c) selecting only spectra of high signal to noise ratio.

The different aspects are discussed in more detail below. Since at the WLG station different detector temperatures were used for different time periods, the spectral analysis was performed with different spectral calibrations (and corresponding sets of

10 convoluted cross sections) for each of these periods. Besides that, the spectral analysis was carried out with consistent settings for the different periods. In order to limit the amount of work, only 'long periods' that contained at least 60 measurement days were selected for the data analysis (see Table 1).

Table 1 Selected periods with different (stable) detector temperatures, which were selected for the data analysis

| Period | Number of days |
| --- | --- |
| 01.04.2012 – 30.06.2012 | ~60 days |
| 19.07.2012 – 02.12.2012 | ~130 days |
| 02.12.2012 – 24.06.2013 | ~200 days |
| 24.06.2013 – 21.11.2013 | ~150 days |
| 22.11.2013 – 07.03.2014 | ~100 days |
| 02.04.2014 – 20.05.2014 | ~50 days |
| 20.05.2014 – 01.10.2014 | ~130 days |
| 17.10.2014 – 04.04.2015 | ~170 days |
| **Total** | |
| 01.04.2012 – 04.04.2015 | ~990 days |

### 4.1 Averaging of individual spectra

Averaging of spectra increases the signal to noise ratio and thus reduces the statistical error of the spectral retrieval. In order to achieve a large reduction of the statistical error, as much as possible individual spectra should be averaged. However, besides the statistical errors also systematic errors occur, e.g. caused by imperfect correction of the Ring effect or different

20 saturation levels of the detector. Such systematic errors tend to increase if an increasing number of spectra are averaged, because the solar zenith angle (and other atmospheric or instrumental properties) changes during the selected period. Thus it is important to find an 'optimum number' of individual spectra for the averaging, for which a minimum of the fit error is found.

    In Fig. 3 the effect of averaging of different numbers of spectra on the RMS of the spectral analysis is shown. The

25 analysis of spectra averaged from 10 original spectra leads to a strong reduction of the RMS (decreases by a factor of 2 to 3.5) compared to the results for individual spectra, indicating that for individual spectra the total error is dominated by noise.



However, further averaging of spectra (40 original spectra) only leads to a rather small improvement of the RMS. Thus in this study, averages of 10 original spectra are analysed.

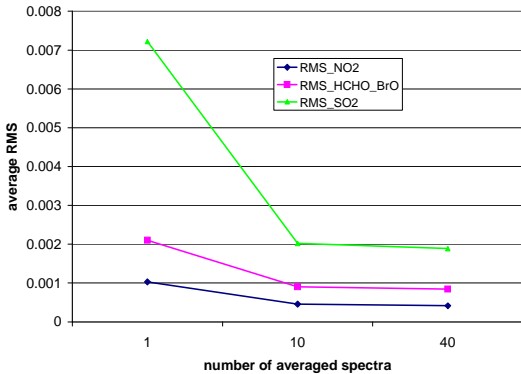

**Figure 3** Mean RMS of the NO₂, SO₂, and HCHO and BrO analysis for spectra taken at 1°, 6°, and 16° elevation for clear sky measurements in April 2013 as function of the number of averaged spectra.

**4.2 Choice of Fraunhofer reference spectra**

Usually, spectra at high elevation angle (usually 90°) and low solar zenith angle (SZA) are used as a Fraunhofer reference

spectrum, because such measurements in general contain the smallest atmospheric absorptions. Such a choice was also first used for the analysis of the WLG measurements. However, it turned out that for zenith spectra larger fit errors occurred than for the spectra at low elevation angles (see Fig. A6 (right) in the appendix). Moreover, also unreasonable results were obtained for the trace gas absorptions and the Ring effect (Fig. A6 left). The main reason for the problems of the zenith spectra is probably that no black tube was mounted in front of the telescope lense. Thus direct sun light can fall on the telescope lense when the

instrument points to zenith (since the instrument is directed towards the North, no direct sun light falls on the telescope for the low elevation angles). Part of the direct sun light will be scattered by the lense onto the fibre bundle and will be added to the 'regular' scattered sun light. If the lense is covered by dirt, the contribution of the direct sun light might be further increased. Here it should be noted that at the high altitude of the measurement site the contribution of direct sun light is substantially enhanced compared to measurements at sea level, because of the reduced molecular scattering in the atmosphere above the

instrument.

The effect of the direct sun light is (at least) twofold:

1) the probability of Raman scattering (Ring effect) will be changed. Thus the correction of the Ring effect will work less good as for spectra of purely scattered sunlight, and spectral interferences with trace gas absorptions might appear. Here it is important to note that unrealistic values for 90° measurements were not only found for the Ring effect, but also for the

absorptions of BrO, HCHO, NO₂ and SO₂.

2) the broad band spectral shape of the spectra changes, the spectra become more 'reddish'. Thus spectrograph straylight will probably be enhanced compared to spectra or purely scattered sun light. Indeed, a much higher variation of the fitted intensity offset is found for spectra in 90° elevation than for the other elevation angles.

In Fig. A6 (left) in the appendix the measured Ring effect (expressed as Raman scattering probability, RSP) is compared to

simulation results of a radiative transfer model. The measurements were analysed using individual Fraunhofer reference spectra taken at 26° elevation of each elevation sequence. Accordingly, also for the simulated RSP values, the corresponding results for 26° elevation were subtracted. The simulations were performed for SZA between 20° and 60° and relative azimuth angles between 60° and 180°, which corresponds to the variation of both quantities for the selected measurements (02 December 2012 to 24 June 2013). The rather large scatter of the measured and simulated RSP values is mainly caused by the





variations of these two quantities. In spite of the rather large scatter, still a large discrepancy between the measured and simulated RSP values is found for 90° elevation, while for the low elevation angles, the agreement is much better. Because of these findings, in this study no measurements at 90° elevation are used as Fraunhofer reference spectra. Instead, individual measurements at 26° elevation of each elevation sequence are used as Fraunhofer reference spectra.

### 4.3 Choice of spectral ranges for the different spectral analyses

The settings for the different spectral analyses are summarised in Table 2. Here it should be noted that the spectral ranges for the retrieval of the different trace gases were determined in dedicated sensitivity studies (see appendix A4). Examples of the spectral analyses are shown in Fig. 4. The errors of the retrieved differential slant column densities (dSCDs) of the trace gas

10 were also estimated based on the sensitivity studies described in appendix A4. They are summarised in Table 3. For all trace gases, the overall error for individual measurements (averages of 10 original spectra) is dominated by random errors. These errors, however, become much smaller if a large number of measurements are averaged.

Table 2 Fit settings for the different spectral analyses.

| BrO / HCHO analysis | |
|---|---|
| Wavelength range (nm) | 314 – 358 |
| DOAS polynomial | degree: 8 |
| Intensity offset | degree: 2 |
| Gaps (nm) | 331.4 – 331.6, 336.4 – 336.8, 349.0 – 349.3 |
| Ring effect | Original and wavelength-dependent Ring spectrum |
| BrO | 228K, Wilmouth et al., 1999 |
| HCHO | 298K, Meller and Moortgart, 2000 |
| $NO_2$ | 220 K, Vandaele et al., 2002 |
| $O_4$ | 293 K, Thalman and Volkamer, 2013 |
| $O_3$ | 223 K, Io corrected, Bogumil et al., 2003 |
| RMS filter | 9e-4 |

| $SO_2$ analysis | |
|---|---|
| Wavelength range (nm) | 306 – 325 |
| DOAS polynomial | degree: 6 |
| Intensity offset | degree: 2 |
| Gaps (nm) | - |
| Ring effect | Original and wavelength-dependent Ring spectrum |





| | |
|---|---|
| SO$_2$ | 273 K, Bogumil et al., 2003 |
| NO$_2$ | 220 K, Vandaele et al., 2002 |
| O$_3$ | 223 K, Io corrected, Bogumil et al., 2003 |
| O$_3$, wavelength dependent | O$_3$ cross section multiplied with wavelength, orthogonalised to original O$_3$ cross section |
| RMS filter | 1.8e-3 |

| **NO$_2$ analysis** | |
|---|---|
| Wavelength range (nm) | 399 – 426 |
| DOAS polynomial | degree: 4 |
| Intensity offset | degree: 2 |
| Gaps (nm) | 422.4 – 423.1 |
| Ring effect | Original and wavelength-dependent Ring spectrum |
| NO$_2$ | 220 K, Vandaele et al., 2002 |
| O$_3$ | 223 K, Io corrected, Bogumil et al., 2003 |
| RMS filter | 5e-4 |

| **O$_4$ analysis** | |
|---|---|
| Wavelength range (nm) | 352 – 387 |
| DOAS polynomial | degree: 5 |
| Intensity offset | degree: 2 |
| Gaps (nm) | - |
| Ring effect | Original and wavelength-dependent Ring spectrum |
| O$_4$ | 293 K, Thalman and Volkamer, 2013 |
| NO$_2$ | 220 K, Vandaele et al., 2002 |
| O$_3$ | 223 K, Io corrected, Bogumil et al., 2003 |





Table 3 Overview on the different systematic and random errors for the retrieved trace gas dSCDs (for spectra averaged from 10 original spectra). More details are found in the appendix.

| error type | BrO | HCHO | $SO_2$ | $NO_2$ |
|---|---|---|---|---|
| systematic | 3e12 molec/cm² | 4e15 molec/cm² | 8e15 molec/cm² | 6e14 molec/cm² |
| random | 1.1e13 molec/cm² | 9e15 molec/cm² | 1.3e16 molec/cm² | 2e15 molec/cm² |

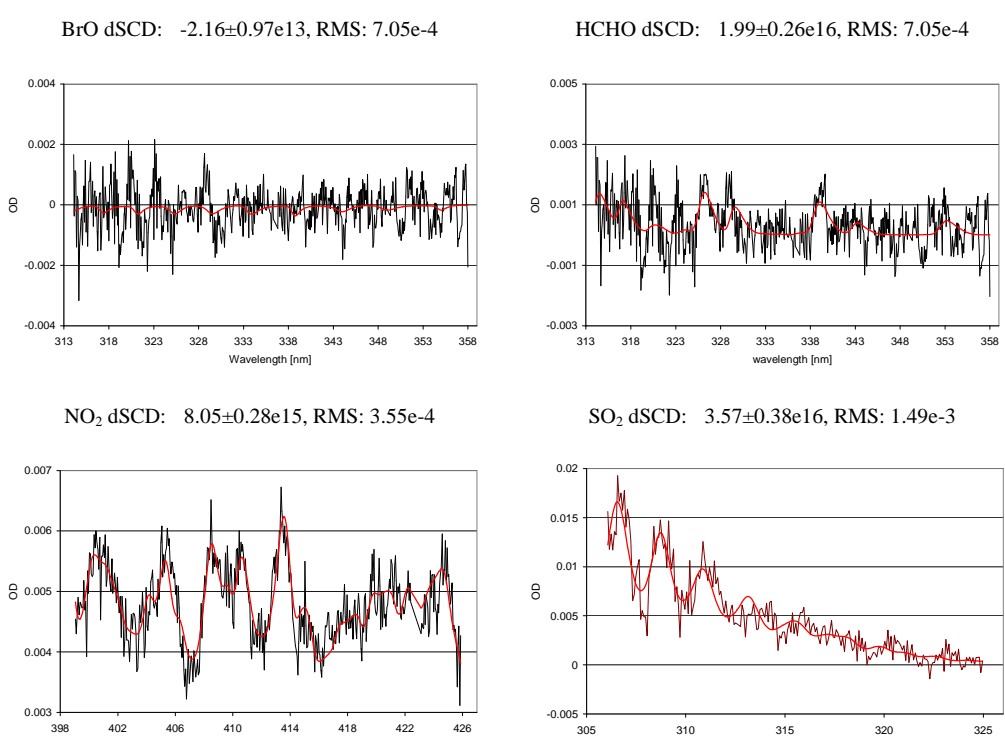

**Figure 4** Fit result for BrO, HCHO, $NO_2$, and $SO_2$ for a spectrum (average of 10 original spectra) taken on 13 May 2013 (00:34 – 04:19) at 1° elevation. On this day, enhanced absorptions of $SO_2$ and $NO_2$ were observed.

## 5 Radiative transfer simulations

For the radiative transfer simulations the RTM TRACY-2 was used (Wagner et al., 2007b). This model allows to explicitly consider the variation of the topography around the measurement station. This option was, however, only used in one dimension (in viewing direction) in order to minimise the computational effort. The variation of the surface terrain height and the results of the radiative transfer simulations are illustrated in Fig. 5.

The surface albedo was set to 7.5%. Sensitivity studies indicated that the exact choice is not critical. The aerosol extinction was varied, but was assumed to be constant between 2600 and 5600m altitude. Different aerosol loads (AOD between 0 and 0.5) were assumed. Here it should be noted that only a fraction of 60% of the total AOD is located above the instrument.

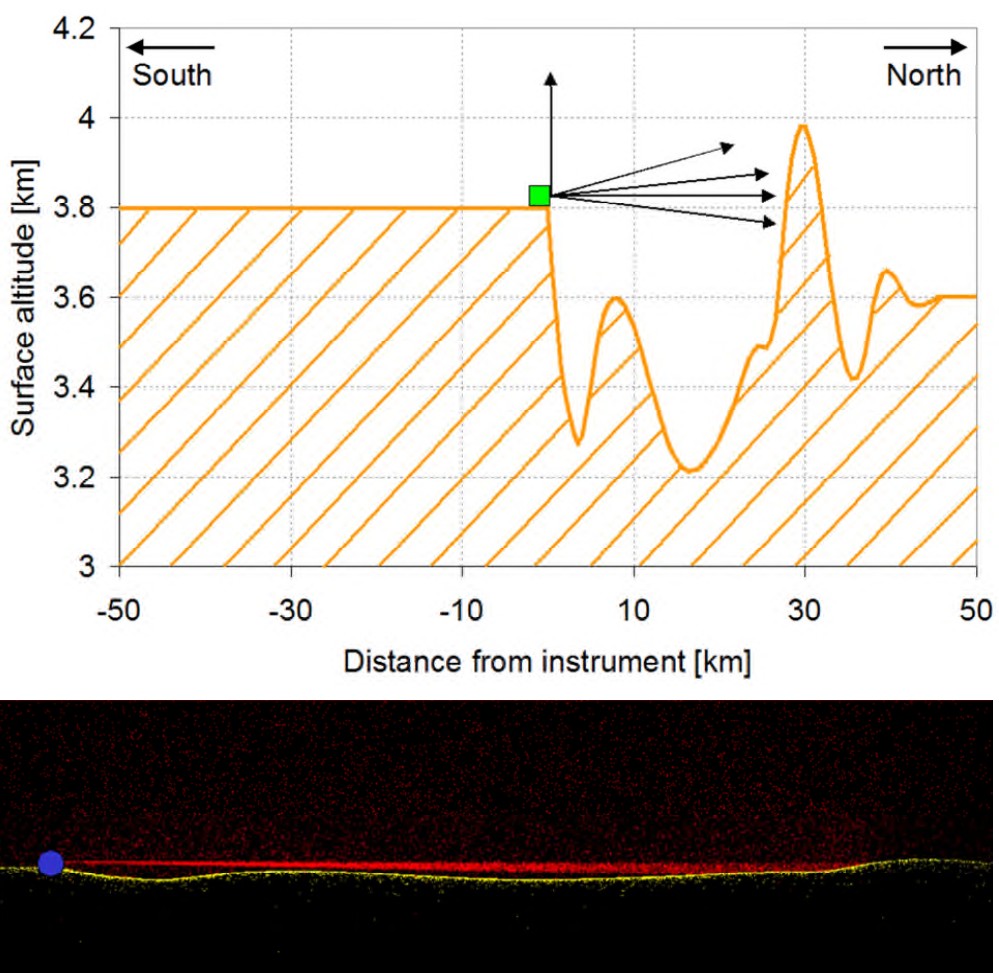

**Figure 5 Top:** Variation of the surface altitude in viewing direction (towards the North). The variation of the surface altitude
across the viewing direction was not explicitly considered to minimise the computational effort. For the same reason, also the
topography 'behind' the instrument, towards the South was assumed to be flat. **Bottom:** Illustration of the results of the
radiative transfer simulations for the area between the instrument and the high mountain in 30 km distance. The blue dot
indicates the position of the instrument. The small red and yellow dots indicate Rayleigh-scattering events and surface
reflection of the simulated solar photons, respectively.

Simulations of trace gas air mass factors (AMFs) were performed for specific viewing geometries for a whole diurnal
cycle in January and July. These months were chosen, because they represent the most extreme viewing geometries (winter
and summer) during the whole year. From the derived diurnal variations of the trace gas AMFs, daily averages for
measurements with SZA below 65° were calculated. For these calculations the individual AMFs were weighted by the
corresponding simulated intensities. Finally, the simulated AMFs for 26° elevation angle were subtracted from the AMFs for
the lower elevation angles yielding the respective differential AMFs (dAMFs). This procedure was applied in order to
calculated trace gas dAMFs which can be directly compared to the trace gas dSCDs derived from the measurements.





### 5.1 Input trace gas profiles for the simulations

In order to relate the measured trace gas dSCDs to atmospheric trace gas mixing ratios, assumptions about the vertical distributions of the trace gases have to be made. The assumed trace gas profiles are described in the following sub sections. Two types of input profiles are used: For the first group of trace gases, the influence of the stratospheric absorptions can be neglected. This is the case for $SO_2$ and HCHO, for which the stratospheric amounts (except for strong volcanic eruptions) are very small and can be neglected. Although for $NO_2$, the contribution from the stratospheric absorption can be rather large, it is found that the stratospheric $NO_2$ absorptions are very similar for the different elevation angles (see appendix A5.1), and the stratospheric absorptions almost completely cancel out for the derived trace gas dSCDs using sequential Fraunhofer reference spectra. Thus, the tropospheric partial dSCD can be simulated independently from the stratospheric absorptions and can be directly compared to the measured $NO_2$ dSCDs.

For BrO, the situation is different: since the stratospheric BrO profile is located at rather low altitudes, the corresponding absorptions depend substantially on the elevation angle. Thus they don't cancel out in the retrieved BrO dSCDs. In fact, the dSCDs for low elevation angles even can become negative (see appendix A5.2) due to the stratospheric BrO absorptions. Therefore, for BrO the stratospheric and tropospheric profiles of BrO have always to be considered simultaneously in the radiative transfer simulations.

### 5.1.1 $SO_2$, $NO_2$ and HCHO

For these trace gases vertical concentration profiles are determined assuming a constant mixing ratio throughout the atmosphere. Of course this is a rather strong simplification of the true profiles, which usually have much more complex shapes. But based on this simple assumption it is still possible to estimate the approximate tropospheric trace gas mixing ratios of $SO_2$, $NO_2$, and HCHO from the corresponding measured trace gas dSCDs. In Fig. 6 the trace gas concentration profiles used in the RTM simulations are shown. They are calculated for typical background mixing ratios of the trace gases. However, it should be noted that the exact knowledge of the true mixing ratio is not important, because for these weak atmospheric absorbers, the air mass factors are almost independent from the absolute trace gas concentrations.





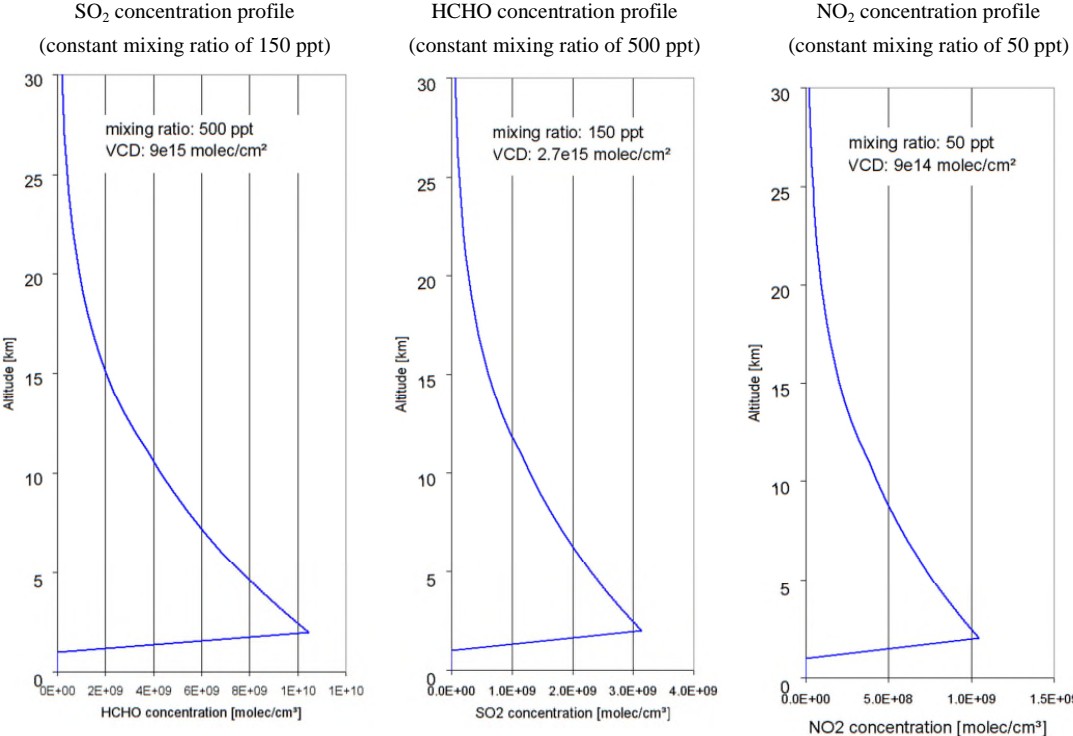

**Figure 6** Vertical concentration profiles of $NO_2$, $SO_2$, and HCHO used in the radiative transfer simulations. In the figures also the corresponding vertical column densities (VCDs) are given.

In Fig. 7 the trace gas dSCDs of $SO_2$, $NO_2$, and HCHO corresponding to the used input profiles are shown as a function of the elevation angle. The trace gas dSCDs are given for different aerosol loads and for different seasons (left: summer; right: winter). Interestingly, for HCHO and $SO_2$, the dSCDs at 1° elevation angle are almost independent from the aerosols load. In contrast, for the $NO_2$ dSCDs at 1° elevation angle a larger dependence on the aerosol load is found, probably because of less Rayleigh scattering at these longer wavelengths. Nevertheless, for simulations with AODs between 0.1 and 0.5, also the $NO_2$ dSCDs at 1° elevation angle very similar.

From the simulations results for 1° elevation angle (and low aerosol load: AOD = 0.1) approximate relationships between the trace gas dSCDs (at 1° elevation angle) and the corresponding volume mixing ratios are derived. They are given below for the different trace gases:

**$SO_2$:** a dSCD of $1 \times 10^{15}$ molec/cm² corresponds to a mixing ratio of 60 ppt.

**$NO_2$:** a dSCD of $1 \times 10^{15}$ molec/cm² corresponds to a mixing ratio of 23 ppt.

**HCHO:** a dSCD of $1 \times 10^{15}$ molec/cm² corresponds to a mixing ratio of 42 ppt. It should be noted that while in the simulations a constant mixing ratio throughout the atmosphere was assumed, the measured trace gas dSCDs are mainly sensitive to the trace gas concentrations in the atmospheric layers close to the instrument. Thus the derived trace gas mixing ratios are most representative for the free troposphere in the altitude range between about 4 and 5 km.

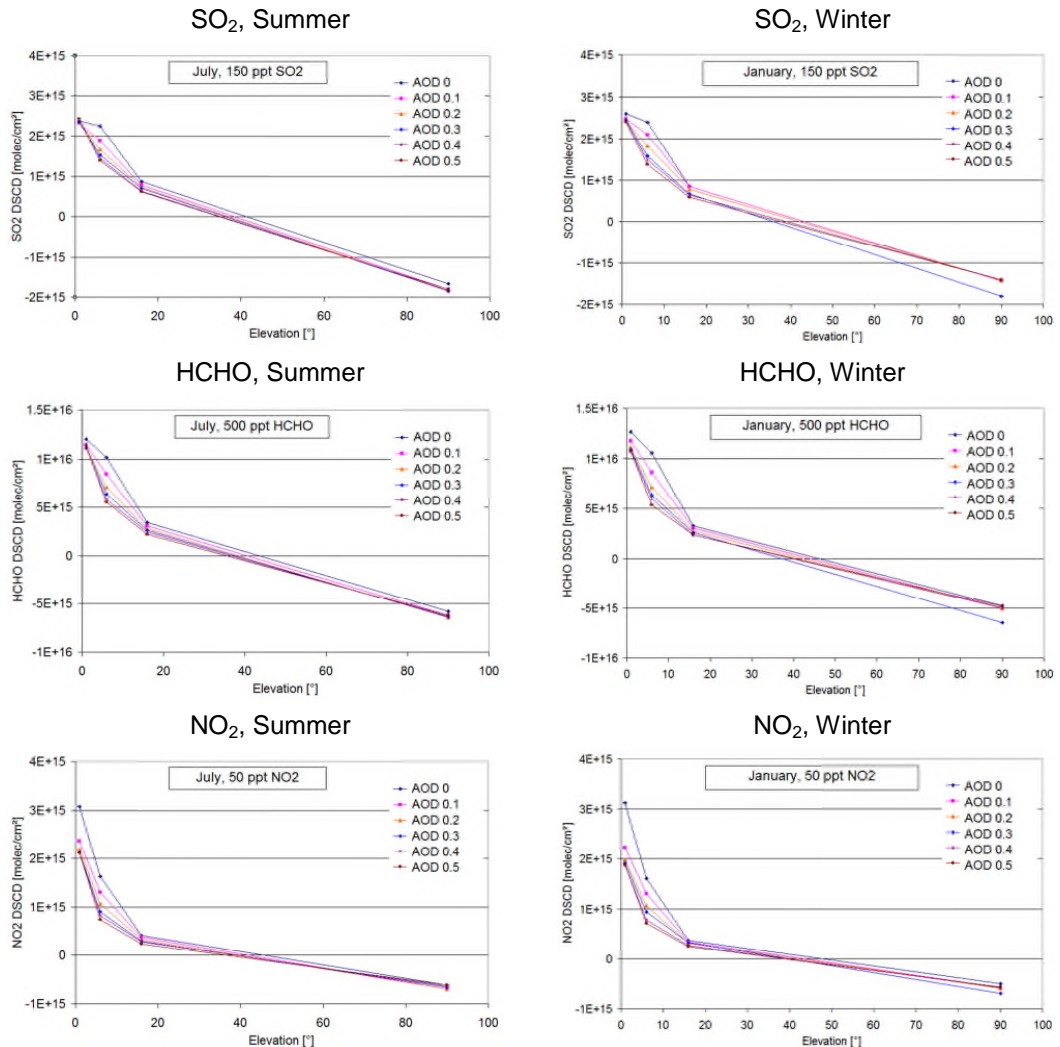

**Figure 7** Trace gas dSCDs simulated for the profiles shown in Fig. 6 for summer (left) and winter (right). The individual lines represent results for different aerosol loads.

### 5.1.2 BrO

For BrO the effect of the stratospheric BrO cannot be separated from the effect of (possible) tropospheric BrO absorptions. This is caused by the fact that the maximum of the stratospheric BrO is usually found only a few kilometres above the tropopause. Thus, the BrO dSCDs retrieved for the stratospheric BrO absorption depend systematically on the elevation angle. Moreover, variations of the tropopause height directly influence the stratospheric BrO profile. Thus the measured BrO dSCDs become systematically dependent also on the tropopause height. Dort et al. (2008) investigated the dependence of the stratospheric BrO mixing ratios as a function of the relative altitude with respect to the tropopause height (see appendix A5.2 and Fig. A21). The mixing ratio profile presented in Dorf et al. (2008) was also used in this study to calculate BrO concentration profiles as a function of the tropopause height (see Fig. 8). Here it should be noted that for the maximum BrO mixing ratio a slightly lower value (15 ppt) was used as in Dorf et al. (2008) (16 ppt). This reflects the decrease of the stratospheric BrO load in the period between the measurements used in Dorf et l. (2008) and the measurements used in our





study. The corresponding BrO dSCDs are shown in Figs A22 and A23. Interestingly, negative BrO dSCDs are found for measurements at 1° elevation angle if no BrO was assumed in the troposphere (Fig. A22). Substantially higher BrO dSCDs are found for the cases when a tropospheric background mixing ratio of 1 ppt is assumed (Fig. A23).

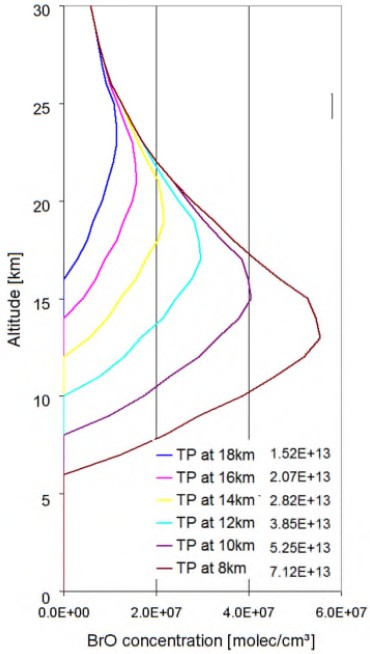
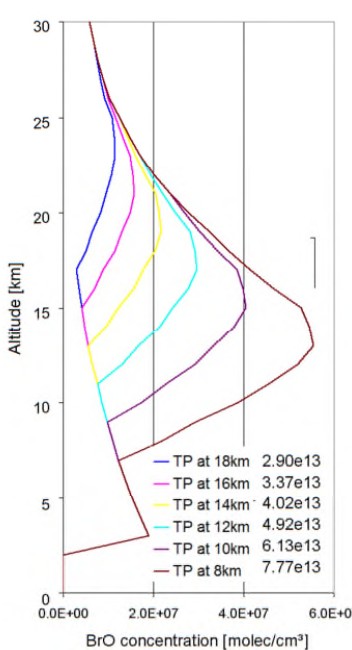

**Figure 8** BrO concentration profiles as function of the tropopause height. **Left:** no BrO in the troposphere; **right:** 1ppt BrO in the troposphere (above 3km). The numbers in the figures represent the corresponding BrO VCDs.

**6 Identification of measurements made under cloudy sky conditions and high aerosol loads**

For the quantitative interpretation of the measurements, they are compared to results from radiative transfer simulations.
These simulations are performed for well defined, in particular cloud-free conditions. Thus only measurements for such conditions have to be selected. Moreover, to make benefit of the high sensitivity of the MAX-DOAS measurements, situations with low aerosol load and thus high visibility have to be selected. The following two sub-sections describe how measurements under cloudy conditions and high aerosol loads are identified.

**6.1 Cloud filter**

Cloudy sky conditions can be identified and classified by different quantities (see e.g. Wagner et al., 2014, 2016; Gielen et al., 2014). In this study, to minimise the computational effort, we only use the colour index (CI) measured in zenith direction. We chose the wavelength pair 330 nm and 390 nm:

$$CI = \frac{signal(330nm)}{signal(390nm)}$$

(1)



Moreover, in order to minimise the potential effects of instrument degradation, not the absolute value of the CI is used for the cloud classification. Instead, two derived quantities are calculated:

**1) The temporal smoothness indicator (TSI)**

The TSI is derived from the zenith measurements. If the CI between subsequent zenith measurements changes rapidly, this indicates the presence of clouds. The TSI is calculated according to the following formula:

$$TSI_i = \left| \frac{CI_{i-1} - CI_{i+1}}{2} - CI_i \right|$$

(2)

Here i indicates the number of an elevation sequence. For clear sky (and homogenous cloud cover) the TSI is small. For broken clouds the TSI is large.

**2) The spread (SP) of the CI for one elevation sequence**

The SP is calculated as the difference between the maximum and minimum of the CI for a selected elevation sequence:

$$SP_i = \max(CI_i) - \min(CI_i)$$

(3)

Here $\max(CI_i)$ and $\min(CI_i)$ indicate the maximum and minimum CI of the considered elevation sequence. For clear sky, the SP is large, for (homogenous) clouds the SP is low.

Based on the calculated TSI and SP the cloud situation of an individual elevation sequence is classified as clear sky, broken clouds or continuous clouds according to the following thresholds:

    a) Clear sky:          TSI < 0.012 and SP > 0.15,

    b) Broken clouds:       TSI > 0.012,

    c) Continuous clouds : TSI < 0.012 and SP < 0.15.

Note: for cases a) and c) both TSI (at the beginning and the end of the elevation sequence) have to be < 0.012; for case b) the condition is fulfilled if one of both TSI is > 0.012

**6.2 Aerosol filter**

For the low elevation angles the atmospheric visibility and thus the length of the light path depends strongly on the aerosol load. Thus measurements with high aerosol loads have decreased sensitivity to the trace gas absorptions and have thus to be identified and removed from further processing. For that purpose the retrieved $O_4$ absorption is used (Wagner et al., 2004;Lampel et al., 2018). In the following a threshold for the retrieved $O_4$ dSCD at 1° elevation angle of $1.4 \times 10^{43}$ molec²/cm⁵ is used, which corresponds to an $O_4$ dAMF of 1.2 (see Fig. 9). This threshold represents an AOD of about 0.1 at 360 nm.

In Fig. 10 the seasonal variation of the $O_4$ dSCDs at 1° elevation is shown. High values are typically found in winter indicating low AOD. In other seasons smaller $O_4$ dSCDs are found indicating higher AOD. This seasonal dependence is in good agreement with measurements of the AOD (Che et al., 2011).





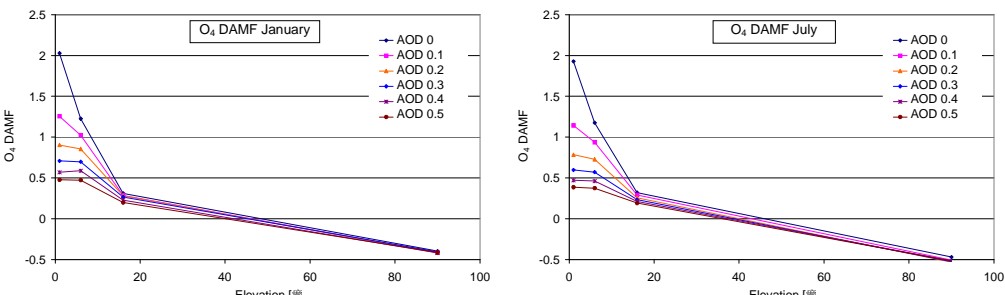

**Figure 9** Simulated $O_4$ dAMFs for different aerosol loads for winter (left) and summer (right). Constant aerosol extinction was assumed between 2600 and 5600m.

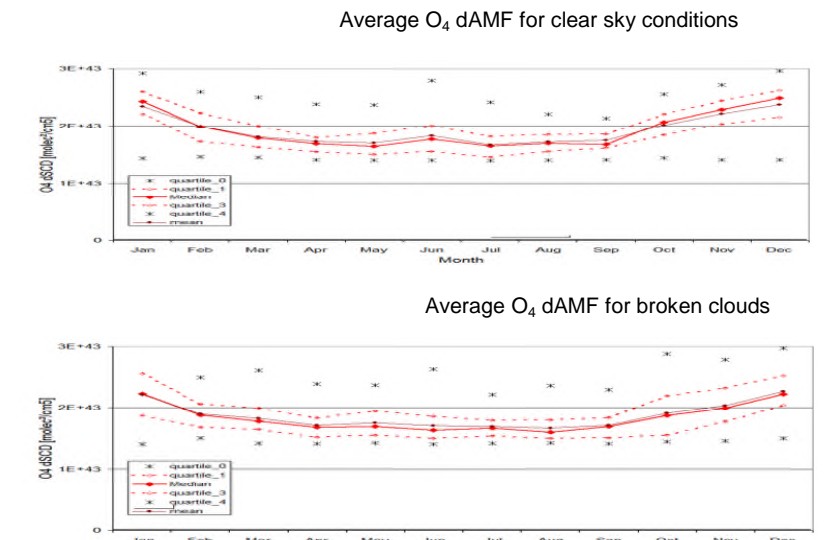

**Figure 10** $O_4$ dSCDs at 1° elevation for clear sky **(top)** and broken clouds **(bottom)** spectra (averages of 10 original spectra for 2012 - 2015) with number of scans > 800 and RMS of the $O_4$ fit < 2e-3.

### 6.3 Summary of sky conditions

In Fig. 11 the seasonal variation of the sky conditions is shown. It is derived by applying the cloud and aerosol classification algorithms described above. The statistics is based on the number of observations (at 1°, 6°, or 16° elevation angle) of

15 spectra averaged from 10 original spectra (April 2012 – April 2015). Only measurements with more than 800 scans are considered. The basic colours indicate the cloud properties (clear, broken clouds, continuous clouds). The full or light colours indicate observations with low or high aerosol loads, respectively.

While the absolute frequency of clear sky observations (low and high aerosol load) stays almost constant over the year, the relative fraction changes strongly with the highest probability of clear sky observations in winter and the lowest

20 probability in summer. The relative fraction of low aerosol cases is largest in winter. The corresponding seasonal frequency plots for the different trace gas analyses (after application of the individual RMS filters) are shown in the appendix A6 (Fig. A24). While the absolute amount of valid data is different for the different analyses, the seasonal frequency is almost the same.





**Number of observations per month**  **Relative fraction per month**

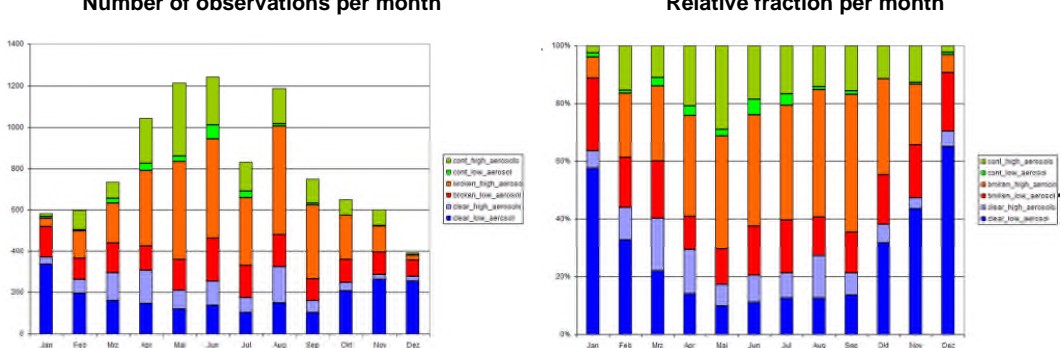

**Figure 11** Absolute **(left)** and relative **(right)** frequency of the different sky conditions. The statistics is based on the number of observations (at 1°, 6°, or 16° elevation angle) of spectra averaged from 10 original spectra (April 2012 – April 2015). Only measurements with more than 800 scans are considered. The basic colours indicate the cloud properties (clear, broken clouds, continuous clouds). The full or light colours indicate observations with low or high aerosol loads, respectively.

## 7 Results

### 7.1 Seasonal means of the dSCDs

In Fig. 12 (left) time series of daily averages of the individual trace gas results for 1° elevation angle are shown. On the right side, the corresponding monthly mean values are shown. In Fig. 13 the seasonal cycles for all elevation angles (1°, 6°, 16°) are shown for clear (left) and broken cloud conditions. In this figure, also the systematic uncertainties of the trace gas dSCDs are indicated by the blue dotted lines. The systematic uncertainties can be regarded as indicators for the lower bounds of the detection limit (which might be reached if a large amount of measurements is averaged). For $NO_2$, $SO_2$, and HCHO, also approximate mixing ratios derived for the measurements at 1° elevation angles are indicated by the y-axes at the right side (see Sect. 5.1.1).

The main findings are:

i) For $NO_2$ and HCHO higher dSCDs are found for lower elevation angles, and this indicates enhanced trace gas mixing ratios in the troposphere (at least in the atmospheric layers between about 4 and 5km);

ii) the highest $NO_2$ values are found in a period from April to June, most likely due to the influence of long range transport of $NO_2$ and its reservoir from both human and natural sources (Ma et al., 2002b;Wang et al., 2006)

iii) for BrO the opposite dependence is found, and this is mainly caused by the influence of stratospheric BrO (see section 5.1.2);

iv) for $SO_2$ no clear elevation dependence is found (the values are below detection limit).





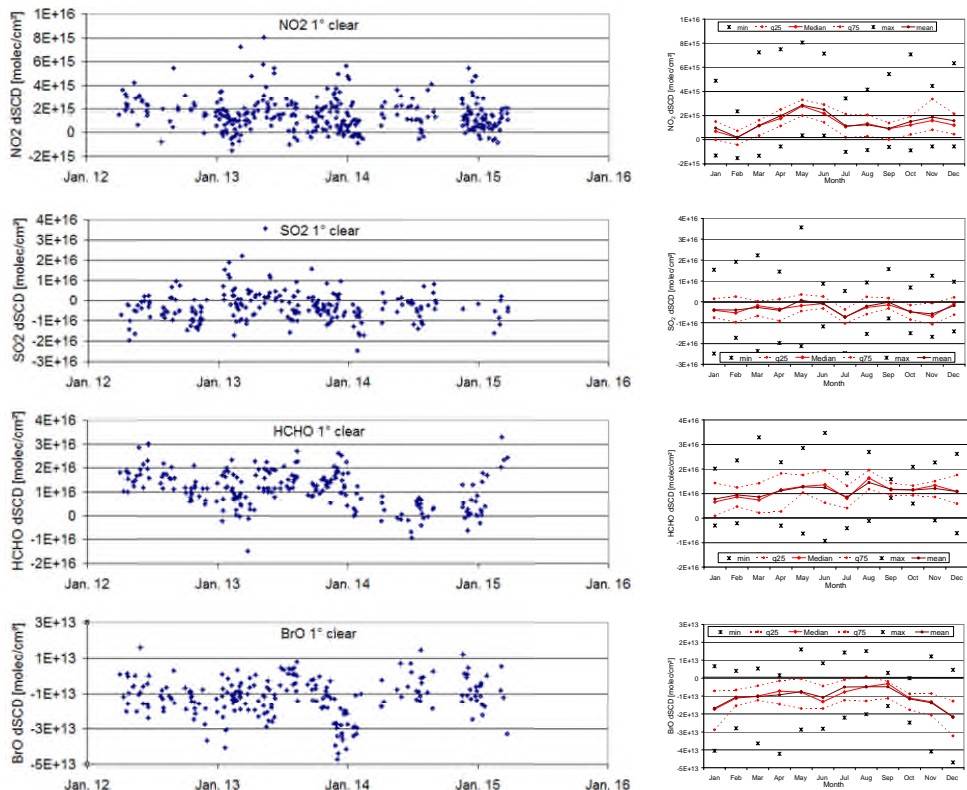

**Figure 12 Left:** Time series of daily averaged trace gas dSCDs at 1° elevation for clear sky spectra and low aerosol load. **Right:** Corresponding seasonal averages.

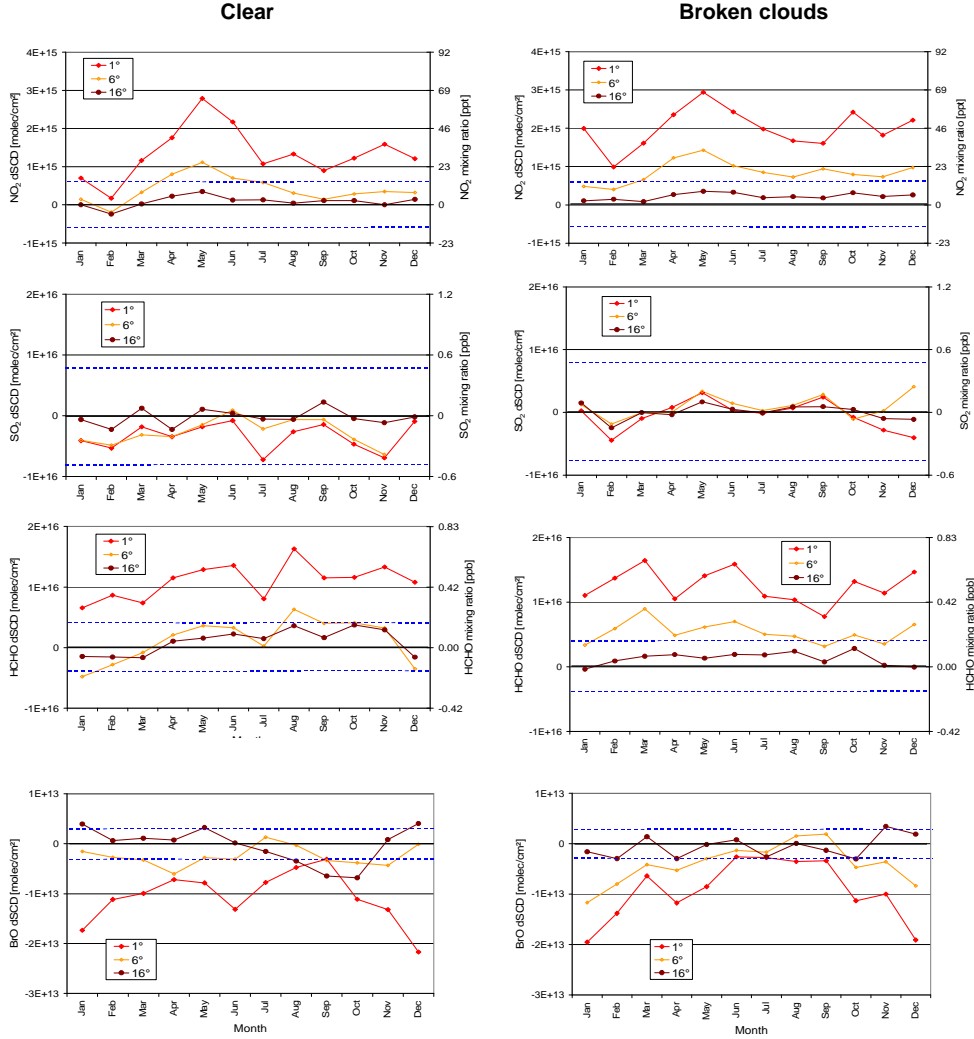

**Figure 13** Seasonal means of the trace gas dSCDs for different elevation angles. **Left:** results for clear sky and low aerosol load, **right:** results for broken clouds and low aerosol load. For $NO_2$, $SO_2$, and HCHO the right axes represent the approximate mixing ratios for measurements at 1° elevation angle. The blue dotted lines indicate the systematic uncertainties, which can be considered as lower bound of the detection limit.

### 7.2 Correlation of the dSCDs at 1° elevation with meteorological parameters

In Fig. 14 correlation coefficients (r) with different meteorological parameters are shown for daily averaged data. Highest correlations are found for BrO, especially with temperature and the tropopause height (r>0.35). Note that temperature and tropopause height are themselves highly correlated (r=0.74), see Fig. A5. For the other trace gases, only low correlations are found with temperature and tropopause height (|r|<0.2).

For $NO_2$ a slight positive correlation with the wind direction below 8km is found, indicating a potential influence of transport from nearby emission sources, and possibly also long range transport (Wang et al., 2006). Also for BrO a slight positive correlation with the wind direction below 5km, and a slight negative correlation with the wind direction between 5km and 8km is found. The interpretation of this finding is not very clear. However, the positive correlation for wind





directions below 5km might be a result of the large positive correlation between the temperature (and tropopause height) with the wind directions below 5km (see Fig. A5). For all trace gases a negative correlation with the wind speeds at all altitudes is found. For $NO_2$, $SO_2$, and HCHO this might simply indicate a dilution effect. For BrO it might be the result of the large negative correlation between the temperature (and tropopause height) with the wind speeds at different altitudes (see

Fig. A5).

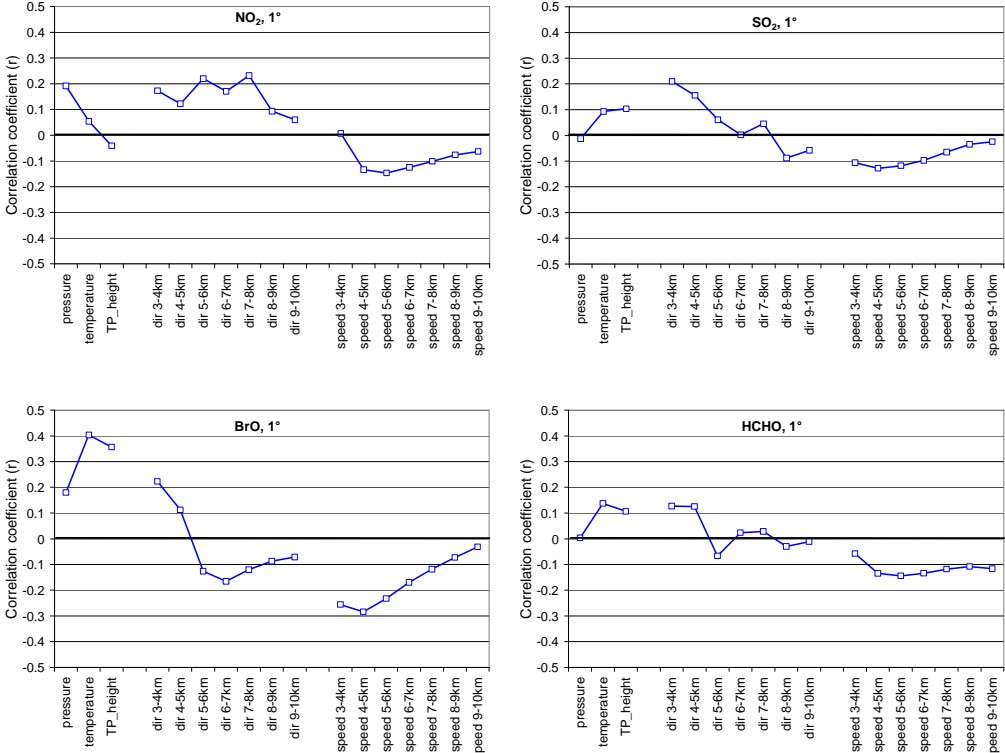

**Figure 14** Correlation coefficients (r) of the daily trace gas dSCDs at 1° elevation angle with different meteorological parameters.

The dependencies on wind speed and wind direction are further investigated by averaging the trace gas dSCDs for different bins of wind speed and direction at selected altitude layers (see Figs. A25 and A26 in the appendix). For $NO_2$, $SO_2$ and BrO systematic dependencies are found for the yearly averaged data, at least for some height ranges. For $NO_2$ and $SO_2$ these dependencies might indicate the effect of long range transport. However, the wind speeds at these layers are also correlated with the seasonal variation of surface temperature and the tropopause height. The latter probably explains the

observed correlation with BrO (see also plots for the dependencies for different season). For HCHO no clear dependence on wind speed and wind direction is found. Overall, the dependencies for individual seasons are not very clear because of the bad statistics.

    The correlation of the BrO dSCDs with tropopause height is investigated in more detail in Fig. 15. For 1° elevation angle, a clear increase of the BrO dSCD with tropopause height is found for both measurements and simulations. For the other

elevation angles, the BrO dSCDs show no systematic dependence on the tropospause height. The comparison between measurements and simulations indicates that no enhanced BrO concentration exists in the troposphere.





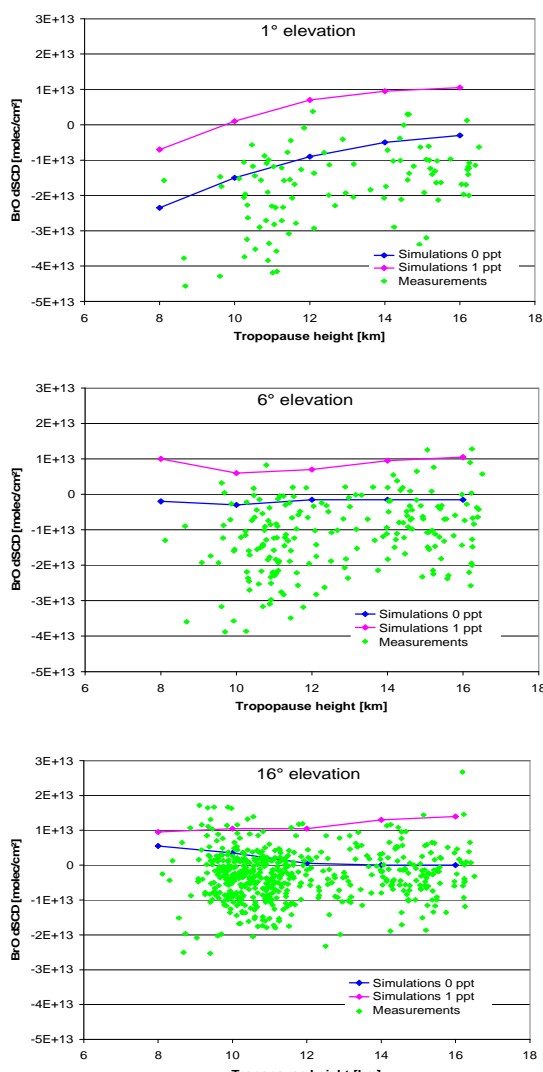

**Figure 15** Comparison of measured and simulated BrO dSCDs as function of the tropopause height for clear sky conditions. The blue and magenta lines represent simulation results for 1 ppt and 0 ppt BrO in the troposphere, respectively.

### 7.3 Estimation of a free tropospheric background mixing ratio from the dSCDs

In this section the tropospheric mixing ratios of the $NO_2$, $SO_2$, and HCHO are estimated based on the respective dSCDs at 1° (see Fig. 13) and the relationships between the dSCDs and the tropospheric mixing ratios (section 5.1.1). For BrO, the tropospheric mixing ratios are estimated based on the direct comparison of the measured and simulated dSCDs, see Fig. 15.

Below are our estimates:

$NO_2$: dSCDs at 1° between $0.2e15 molec/cm²$ (winter) and $3 \times 10^{15}$ molec/cm² (summer) correspond to mixing ratios between about 5 (winter) and 70 ppt (summer).

$SO_2$: dSCDs at 1° below $8 \times 10^{15}$ molec/cm² correspond to mixing ratios below 0.5 ppb,

HCHO: dSCDs at 1° between $0.7 \times 10^{15}$ molec/cm² and $1.7 \times 10^{15}$ molec/cm² correspond to mixing ratios between about 0.3 and 0.7 ppb,


**BrO:** dSCDs at 1° are mostly below the simulation results for BrO profiles without BrO in the troposphere indicating tropospheric mixing ratio ~0 ppt.

It should again be noted that while in the simulations a constant mixing ratio throughout the atmosphere was assumed, the measured trace gas dSCDs are mainly sensitive to the trace gas concentrations in the atmospheric layers close to the instrument. Thus the derived trace gas mixing ratios are most representative for the free troposphere in the altitude range between about 4 and 5 km.

## 7 Conclusions

We made long-term ground-based MAX-DOAS measurements at the WLG WMO/GAW global baseline station during the years 2012–2015. For this study we analyzed the measured spectra to estimate the tropospheric background mixing ratios of different trace gases, including $NO_2$, $SO_2$, HCHO and BrO, from MAX-DOAS measurements at WLG.

For the spectral retrieval, we find that averaging of spectra increases the signal to noise ratio and thus reduces the statistical error of the spectral retrieval on one hand, and systematic errors caused by imperfect correction of the Ring effect tend to increase if an increasing number of spectra is averaged on other hand. Averages of 10 original spectra have been approved to be an 'optimum option' in the spectral analysis for this study. We determined the settings and spectral ranges for the retrieval of the different trace gases by a large number of dedicated sensitivity studies. We performed radiative transfer simulations with the RTM TRACY-2, which allows to explicitly consider the variation of the topography around the measurement station in viewing direction. From the simulations results, approximate relationships between the trace gas dSCDs (at 1° elevation angle and low aerosol load: AOD = 0.1) and the corresponding volume mixing ratios are derived. We used the temporal variation and the spread of the colour index (CI) derived from our MAX-DOAS measurements to select measurement data for clear sky and low aerosol load, and then retrieved the corresponding daily averages and seasonal cycles of the trace gas dSCDs at elevation angles of 1°, 6°, 16°.

For $NO_2$ and HCHO higher dSCDs are found for lower elevation angles, indicating enhanced trace gas mixing ratios in the lower troposphere over WLG. For BrO the opposite dependence is found, reflecting the influence of stratospheric BrO. For $SO_2$ no clear elevation dependence is found. The highest $NO_2$ dSCDs are found in a period from April to June, most likely due to the long range transport of $NO_2$ and its reservoirs to WLG. From the dSCDs at 1° elevation, mixing ratios of $NO_2$ in the lower troposphere over WLG are estimated to be between about 5 ppt (winter) and 70 ppt (summer), and mixing ratios of $SO_2$ fall below 0.5 ppb. Mixing ratios of HCHO range between about 0.3 and 0.7 ppb. These mixing ratios are most representative for atmospheric layers between about 4 and 5 km. Mixing ratios of BrO are estimated to be close to ~0 ppt as its dSCDs mostly correspond to the simulation results without BrO in the troposphere. Since stratospheric BrO is located at rather low altitudes, the corresponding absorptions depend substantially on the elevation angle and thus cannot cancel out in the retrieved BrO dSCDs. Retrieving BrO in the continental background troposphere remains a challenge, which should be further addressed in future studies.





**Appendix**

**A1 Determination and correction of the elevation calibration**

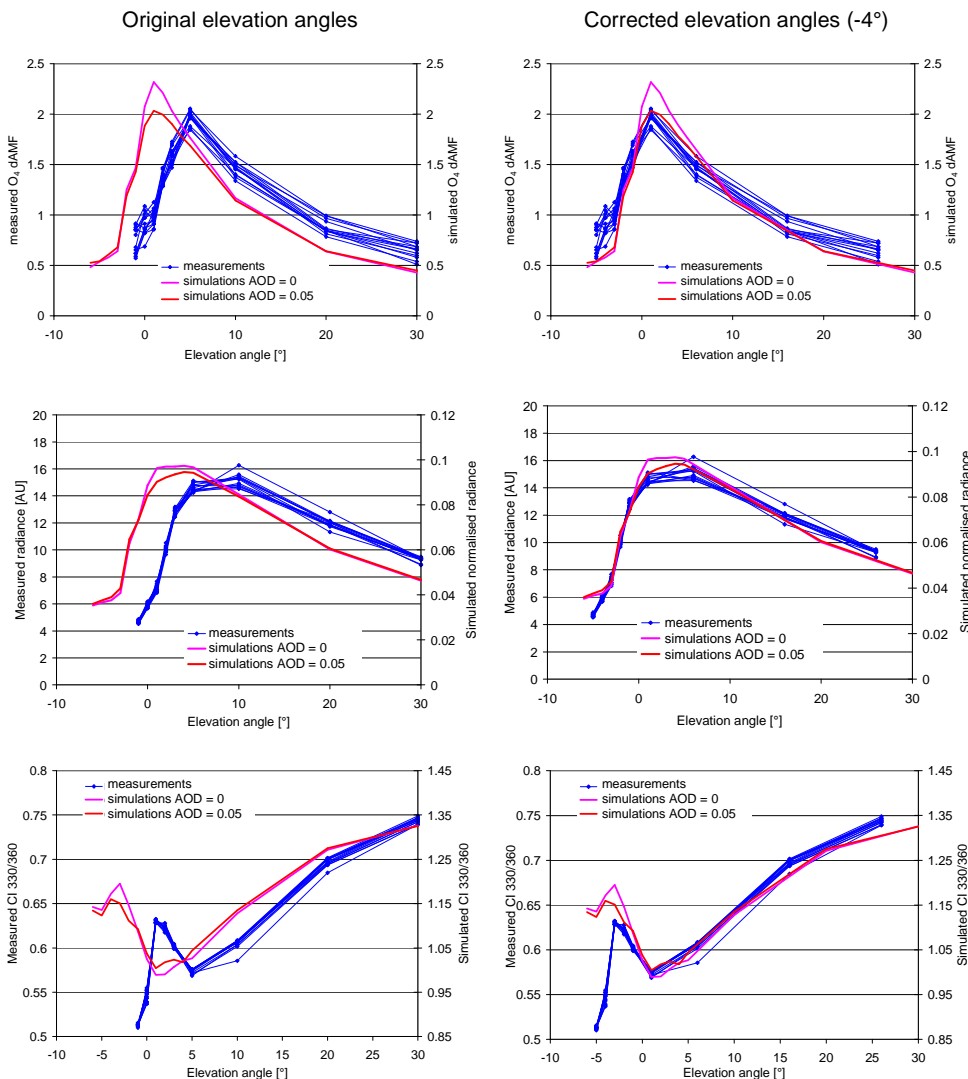

5  **Fig. A1** Comparison of simulated (magenta and red lines) and measured (blue lines) $O_4$ absorption (expressed as $O_4$ air mass factor), radiance at 360 nm, and colour index (CI) for 330 and 360 nm. For the comparison, measurements around noon on a clear day (14.04.2013) were selected. Simulations were made for two different aerosol loads (magenta: no aerosols, red: AOD = 0.05). The SZA is about 30°, and the relative azimuth angle is about 180°. Left: measurements are displayed as function of the original (wrong) elevation angle calibration. The figures at the right show the same data, but with the measurements as

10  function of the corrected (by -4°) elevation angles.





**A2 Time series of additional meteorological quantities; comparison between ECMWF and in situ data; investigation of the correlation between different meteorological quantities**

**Fig. A2** Left: Time series of daily averaged wind speeds at different altitudes above sea level over the WLG station derived
5    from ECMWF data (2012 to 2015). Right: The corresponding seasonal cycles.





**Fig. A3** Left: Time series of daily averaged wind directions at different altitudes above sea level over the WLG station derived from ECMWF data (2012 to 2015). Right: The corresponding seasonal cycles.



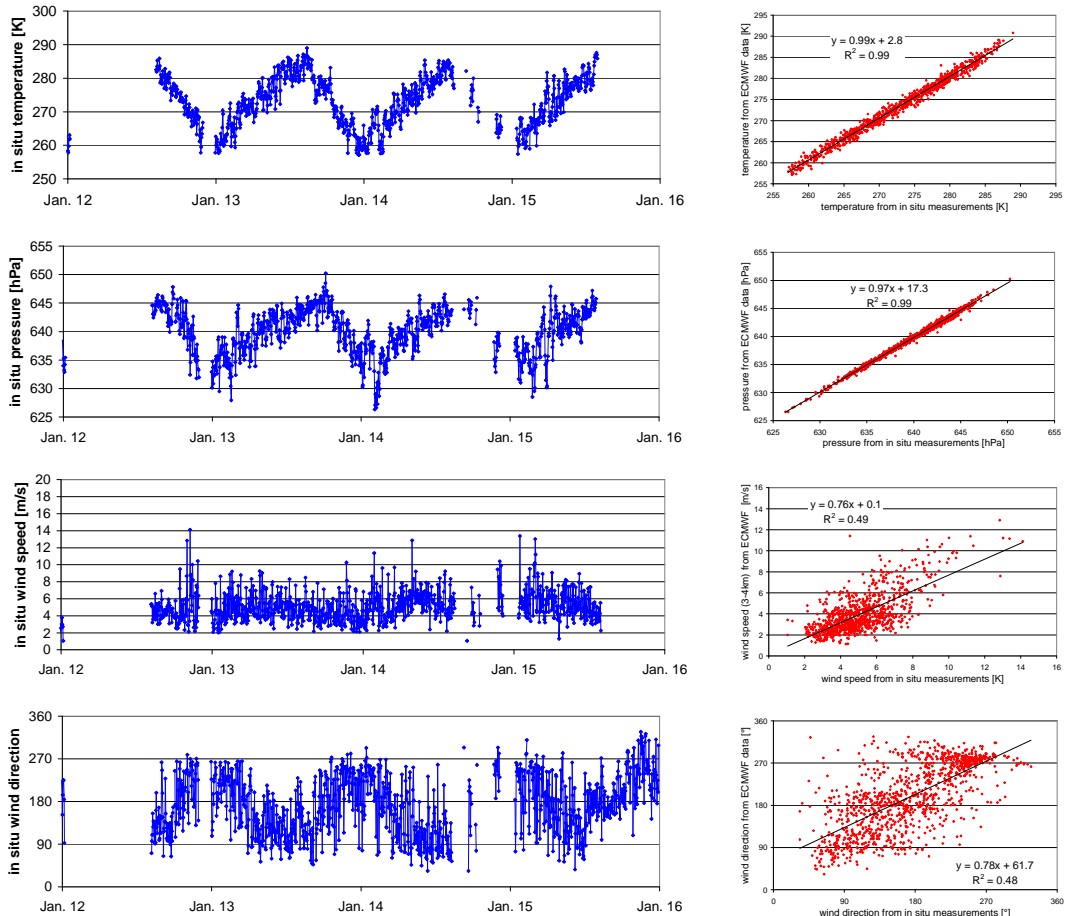

**Fig. A4** Left: Time series of daily averaged meteorological data from the in situ measurements at WLG (2012 to 2015). Right: Correlation plots of the ECMWF versus the corresponding in situ data.





## Monthly averages          Daily averages

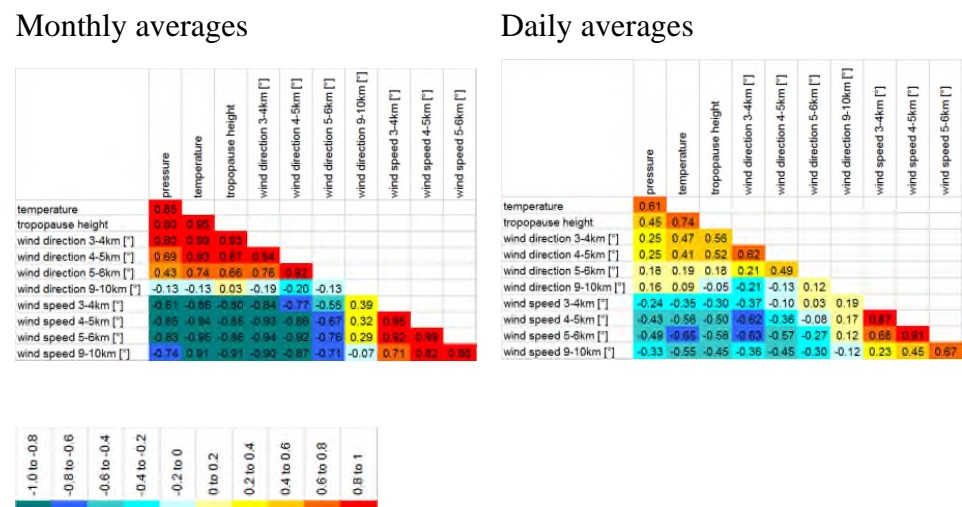

**Fig. A5** Correlation coefficients (r) between different meteorological quantities from ECMWF data (left: monthly averages; right: daily averages).

## A3 Problems for measurements at 90° elevation angle

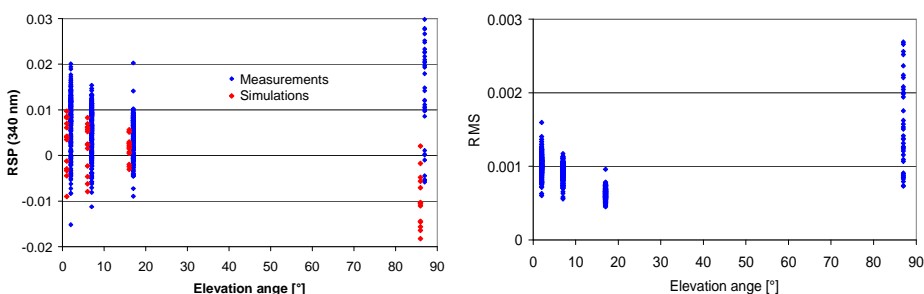

**Fig. A6** Left: Results for the Ring effect (expressed as Raman scattering probability, RSP) as function of the elevation angle for measurements from 02 December 2012 to 24 June 2013 (blue) and derived from radiative transfer simulations (red). Right: RMS of the spectral fit for BrO and HCHO as function of the elevation angle for measurements from 02 December 2012 to 24 June 2013. The measurements were analyzed using Fraunhofer reference spectra taken at 26° of each elevation sequence.


**A4 Determination of fit windows for the different trace gases**

**A4.1 Determination of the optimum fit range for BrO and HCHO**

For this task both synthetic and measured spectra are used. Different tests are performed to find the best suited fit range for both species. The results are summarized in Table A1 below. Based on these results, a fit range from 314 to 358 nm was chosen. The

5   individual tests are described in more detail below.

**Table A1** Best fit ranges based on different test results.

| Test | Optimum lower fit limit for BrO (nm) | Optimum upper fit limit for BrO (nm) | Optimum lower fit limit for HCHO (nm) | Optimum upper fit limit for HCHO (nm) |
|---|---|---|---|---|
| Comparison with input values of synthetic spectra | 314 – 316 | 356 – 360 | 313 – 314 | 356 – 360 |
| Consistency between synthetic and measured spectra | 314 – 316 | 357 – 358 | 314 - 316 | 358 – 359 |
| Fit error (in brackets: results for synthetic spectra) | 312 – 318 (312 – 316) | 357 – 358 (358 – 360) | 312 – 317 | 358 |
| RMS (in brackets: results for synthetic spectra) | 316 – 318 (314 – 318) | 356 – 358 (356 – 360) | 316 – 318 (314 – 318) | 356 – 358 (356 – 360) |
| scatter of results for 1° elevation angle (in brackets: results for synthetic spectra) | 315 – 316 (312 – 313) | 358 – 360 (358 – 360) | 316 (312 – 317) | 358 – 360 (356 – 358) |
| correlation between BrO and HCHO dSCDs for 1° elevation angle (in brackets: results for synthetic spectra) | 312 – 313 (313 – 318) | 358 – 360 (312 – 313) | 312 – 313 (313 – 318) | 358 – 360 (312 – 313) |





| | | | | |
|---|---|---|---|---|
| **Final selection** | **314** | **358** | **314** | **358** |

**Synthetic spectra**

Synthetic spectra were simulated at high spectral resolution for the spectral range 303 – 390 nm using the RTM SCIATRAN. Rotational Raman scattering was included. The simulations were performed for a SZA of 50° and a relative azimuth angle (RAA) of 180°. Surface albedo and altitude were set to 0.07 and 3800m, respectively.

**Table A2** Trace gas cross sections and atmospheric profiles used for the synthetic spectra

| Trace gas | Cross section | Atmospheric profile |
|---|---|---|
| BrO | bro_wil_228_vac.txt | stratospheric profile with maximum at 20 km. VCD: 3e13 molec/cm² |
| HCHO | HCHO_Meller_298_vac.DAT | Box profile in the lowest 1km. VCD: 1e15 molec/cm² |
| NO$_2$ | NO2_vandaele97_220_vac.txt | Box profile in the lowest 0.5km. VCD: 1e15 molec/cm²; stratospheric profile with maximum at 24 km. VCD: 5.22e15 molec/cm² |
| O$_3$ | O3_203K_V3_0.dat<br>O3_223K_V3_0.dat<br>O3_243K_V3_0.dat<br>O3_273K_V3_0.dat<br>O3_293K_V3_0.dat | From the US standard atmosphere: maximum at 22km. VCD: 9.03e18 molec/cm² (337 DU) |
| O$_4$ | o4_thalman_volkamer_293K_corr.xs | O$_4$ derived from temperature and pressure profile |

An aerosol layer between 3800 and 4800m with an AOD of 0.1 was assumed. The single scattering albedo and phase function were chosen according to biomass burning aerosols. For the ozone absorption the temperature dependence was taken into account. Information about the chosen trace gas cross sections and assumed atmospheric profiles is given in Table A2.

The Radiance output is convoluted with a Gaussian function with FWHM of 0.6 nm. Random noise with a RMS of 5e-4 is added to the convoluted spectra. 100 spectra with different noise are simulated for each elevation angle.

In addition to the simulated spectra, also air mass factors are derived from the RTM for the following wavelengths: 315, 340, 355 nm. The resulting dSCDs for BrO and HCHO are shown in Fig. A7. The dSCDs are calculated assuming a Fraunhofer reference spectrum measured at 26° elevation angle. The dSCDs derived in this way are compared to the results of the spectral analyses. For this comparison, the analysis results of the 100 spectra for each elevation angle are averaged. For HCHO and BrO the dSCDs calculated for 340 nm are used for the comparison.





Interestingly, different elevation dependencies are found for both trace gases (Fig. A7). For BrO, the dSCDs decrease towards low elevation angles. This is caused by the fact that in the RTM simulations BrO is only located in the stratosphere. Measurements at 26° elevation (Fraunhofer reference spectra) are more sensitive to these altitudes than measurements at low elevation angles. Thus, for low elevation angles, negative BrO dSCDs are obtained. For HCHO, the opposite elevation dependence is found, because HCHO is only located in the troposphere.

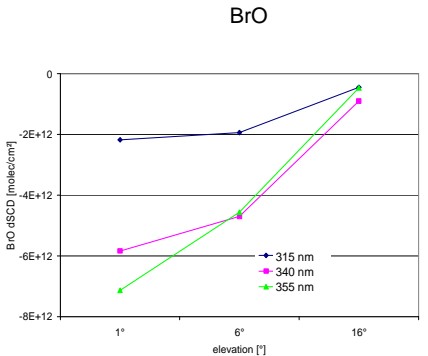
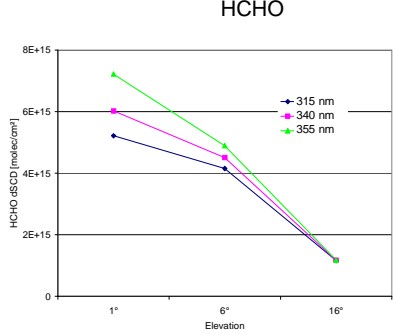

**Fig. A7** Dependence of the simulated trace gas dSCDs for BrO (left) and HCHO (right) on the elevation angle for different wavelengths used for the AMF calculations.

**Measured spectra**

Measurements for the period 02.12.2012 – 24.06.2013 were selected. This period covers ~200 days in different seasons. For the analysis of the measured spectra the same settings as for the synthetic spectra were used. Only measurements for clear sky and with scan numbers > 800 were considered.

**Dependence of the retrieved BrO and HCHO dSCDs on the upper and lower fit boundaries**

Fig. A8 shows results for 1° elevation angle (similar results are found for the other elevation angles). In addition to the results for the measured spectra (red lines), also the results for the synthetic spectra (blue lines) together with the 'true dSCDs' (blue dashed lines) are shown.





**Fig. A8** Dependence of the fit results (at 1° elevation angle) for the measured spectra (red lines) and synthetic spectra (blue lines) on the upper and lower boundaries of the fit range. Also shown are the 'true dSCDs' derived from the simulated air mass





factors (blue dashed lines). The individual plots represent results for one fixed upper wavelength limit. The lower wavelength limit is represented by the x-axes.

The individual figures show the results for one value of the upper limit of the fit range. The x-axes indicate the lower limit of the fit range. For the comparison of the results for measured and synthetic spectra it has to be taken into account that the true atmospheric profiles are not known. They might especially differ from the profiles assumed for the simulation of the synthetic spectra. Thus no perfect quantitative agreement can be expected. However, it is meaningful to compare the overall dependencies on the upper and lower limits of the fit ranges. Here quite good agreement is found (except for the lowest values for the lower fit boundary) indicating that the results for the synthetic spectra are well representative also for the measured

spectra. For BrO the best agreement between the results of the synthetic spectra and the 'true dSCDs' is found for lower limits between 314 and 316 nm, while for the upper limits no clear dependency is found. For HCHO the best agreement between the results of the synthetic spectra and the 'true dSCDs' is found for lower limits between 313 and 316 nm. Again, for the upper limits no clear dependency is found.

**Fit error and RMS**

The errors of the BrO and HCHO dSCDs and RMS values derived from the fit are investigated for the measured and synthetic spectra. While for a fixed wavelength range, an increasing RMS is usually also accompanied by an increasing fit error, both quantities might depend differently on the variation of the fit boundaries. Fig. A9 shows the fit errors and RMS for synthetic (left) and measured spectra (right). In general, systematically higher fit errors and RMS are found for the measured spectra.

This probably indicates higher noise levels, but also additional systematic errors, caused e.g. by imperfect convolution, errors of the input cross sections, or remaining temperature dependencies.



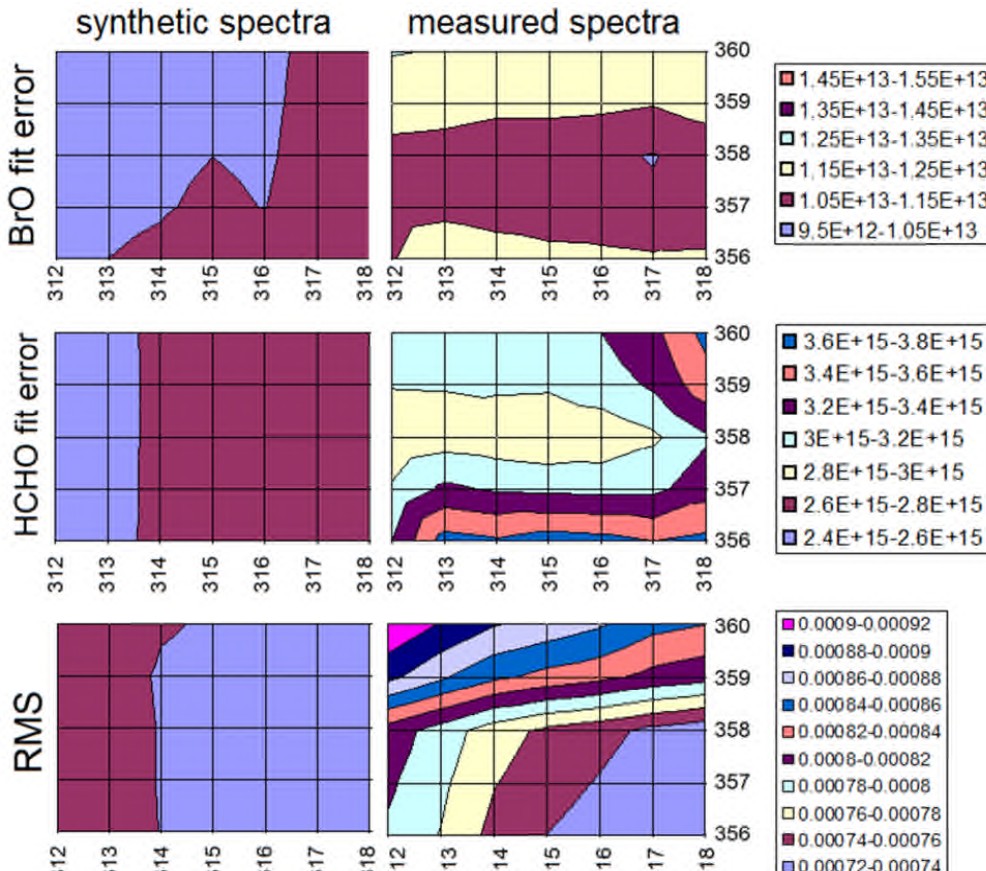

**Fig. A9** Dependence of the fit errors and RMS on the upper and lower boundaries of the fit range for the synthetic (left) and measured spectra (right).

**Standard deviation of the retrieved dSCDs and correlation between the dSCDs of BrO and HCHO**

In this section, the temporal variability of the fit results is investigated. For this task, the results for 1° elevation are chosen, because they are expected to show the highest trace gas dSCDs and the largest variability. Since for the synthetic spectra the same input profiles are used for all spectra, the same trace gas dSCDs should be retrieved. Moreover, no correlation between
10 the dSCDs of BrO and HCHO is expected. However, since noise was added to the synthetic spectra, also the derived trace gas dSCDs might be affected by some random variation, leading eventually also to some correlation between the dSCDs of BrO and HCHO.

For the measurements, also the atmospheric trace gas concentrations can vary. Moreover, also changes of the atmospheric visibility will probably contribute to a variation of the measured trace gas dSCDs. Thus the retrieved variability of the trace gas
15 dSCDs for the measured spectra is expected to be higher than for the synthetic dSCDs. For the measured spectra, a (anti-) correlation of the derived dSCDs of BrO and HCHO might reflect a (anti-) correlation of their true atmospheric absorptions. However, such a (anti-) correlation is not very probable because of the rather different atmospheric profile shapes and





formation processes. Thus a low correlation between the dSCDs of BrO and HCHO is considered as an indication for a good fit quality for both the synthetic and measured spectra.

Figure A10 shows the standard deviation of the time series of the dSCDs of BrO and   HCHO for 1° elevation angle as well as the correlation coefficient between both dSCDs for the synthetic (left) and measured spectra (right). As expected the
5  standard deviations are higher for the measured spectra. Also the correlation between both species is higher for the measured spectra. This finding might partly represent a true correlation of the atmospheric abundances, but is more probably caused by the noise of the spectra.

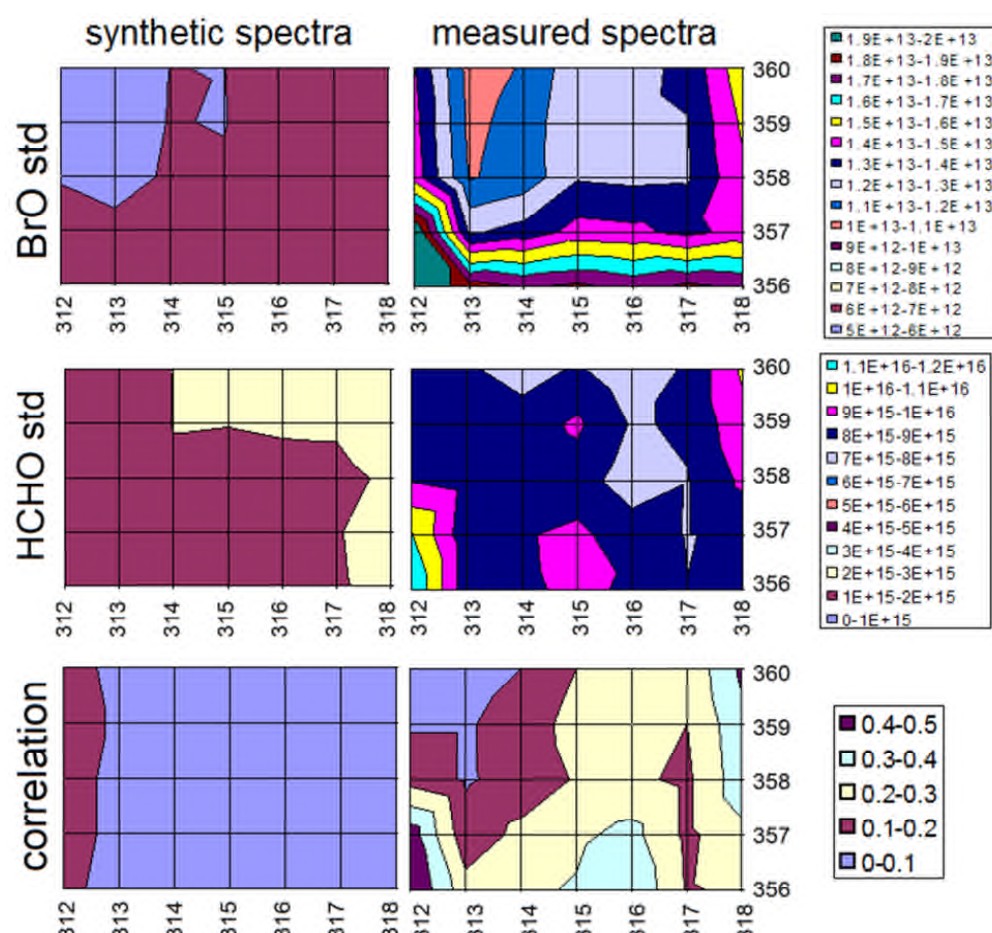

**Fig. A10** Dependence of the variation (expressed as standard deviation) of the fit results for 1° elevation angle and the correlation between the BrO and HCHO dSCDs (expressed as r²) on the upper and lower boundaries of the fit range. For the synthetic (left) and measured spectra (right).





**Final choice of fit the range**

Based on the results from the previous sub-section, the fit range from 314 to 358 nm is chosen for the analysis of BrO and HCHO. This choice is mainly based on the results of the comparison of the retrieved trace gas dSCDS derived from the synthetic spectra to the dSCDs derived from the measured spectra and the 'true dSCDs'. For the chosen fit range, also the fit
errors, the standard deviation of the dSCDs (for 1° elevation), and the correlation between the dSCDs of BrO and HCHO are rather small.

**Error estimate**

The systematic and random uncertainties of the fit results for individual spectra (average of 10 original spectra) are
summarized in Table A3.

**Table A3** overview on the different systematic and random error sources

| error source | error of BrO fit result | error of HCHO fit result |
|---|---|---|
|  |  |  |
| **systematic errors** |  |  |
| Comparison of the fit results for the synthetic spectra to the 'true dSCDs' | 3e12 molec/cm² | 4e15 molec/cm² |
|  |  |  |
| **random errors** |  |  |
| Fit error | 1.1e13 molec/cm² | 3e15 molec/cm² |
| Standard deviation* | 1.1e13 molec/cm² | 9e15 molec/cm² |

*the standard deviation describes the total statistical error; at least part of the total statistical error is caused by the fit error.

The systematic error of the dSCDs is usually dominated by systematic effects of the spectral retrieval, which are quantified by the comparison of the fit results for the synthetic spectra to the 'true dSCDs'. Only for high trace gas dSCDs the uncertainty of the cross section might become important. But for the rather low trace gas dSCDs retrieved in this study the uncertainty of the cross section can be neglected.
        For individual measurements, the random errors clearly dominate the total uncertainty. However, if several measurements
are averaged these errors can be largely reduced.



**Quality filter**

Spectra with bad fit quality have to be removed from further processing. First, measurements with a small number of individual scans (<800) are removed. This filter removes spectra with the worst fit results. In addition, a RMS filter is applied. Here a threshold for the RMS of 9e-4 is chosen. Figure A11 shows the frequency distribution of the RMS (for measurements with more than 800 scans). The RMS threshold removes about 24% of the BrO and HCHO results.

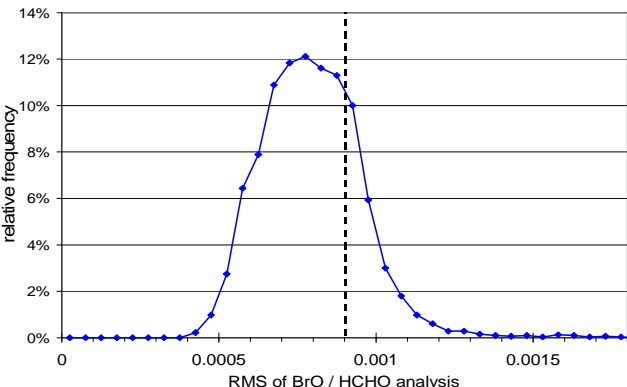

**Fig. A11** frequency distribution of the RMS of the spectral analysis of BrO and HCHO for clear sky conditions. The vertical line indicates the threshold for removing measurements with high fit errors.

**A4.2 Determination of the optimum fit range for SO$_2$**

In this task both synthetic and measured spectra are used. Different tests are performed to find the best suited fit range for both species. The results are summarized in Table A4 below. Based on these results a fit range from 306 to 325 nm was chosen. The individual tests are described in more detail below.

**Table A4** Best fit ranges based on different test results.

| Test | Optimum lower fit limit for SO$_2$ (nm) | Optimum upper fit limit for SO$_2$ (nm) |
|---|---|---|
| Comparison with input values of synthetic spectra | 304 – 308 | 323 – 327 |
| Consistency between synthetic and measured spectra | 306 – 307 | 323 – 327 |
| Fit error | 305 – 306 | 323 – 327 |


| (in brackets: results for synthetic spectra) | (304) | (323 – 327) |
|---|---|---|
| RMS (in brackets: results for synthetic spectra) | 308 (304) | 323 – 327 (323 – 327) |
| scatter of results for 1° elevation angle (in brackets: results for synthetic spectra) | 306 – 307 (304) | 323 – 327 (325 – 327) |
| correlation between BrO and HCHO dSCDs for 1° elevation angle (in brackets: results for synthetic spectra) | 304 & 306 (308) | 323 – 327 (325 – 327) |
| | | |
| **Final selection** | **306** | **325** |

**Synthetic and measured spectra**

The same data sets as for the BrO and HCHO analysis are used also for the determination of the fit range of $SO_2$ (see Sect.
A4.1). Since no $SO_2$ absorptions are included in the simulation of the synthetic spectra, also the derived $SO_2$ dSCDs should be
(close to) zero.

**Dependence of the retrieved $SO_2$ dSCDs on the upper and lower fit boundaries**

Fig. A12 shows results for 1° elevation angle (similar results are found for the other elevation angles). In addition to the results
for the measured spectra (red lines), also the results for the synthetic spectra (blue lines) together with the 'true dSCDs' (blue
dashed lines) are shown. The individual figures show the results for one value of the upper limit of the fit range. The x-axes
indicate the lower limit of the fit range. For the comparison between the results for measured and synthetic spectra it has to be
taken into account that the true atmospheric profiles are not known. They might especially differ from the profiles assumed for
the simulation of the synthetic spectra (zero $SO_2$ absorption). Thus no perfect quantitative agreement can be expected. And
indeed, the derived $SO_2$ dSCDs from the measured spectra differ substantially (by up to 2e16 molec/cm²) from those of the
synthetic spectra. Also the dependence on the lower fit boundary is very different for measured and synthetic spectra. While
the retrieved dSCDs for the synthetic spectra are close to zero, the retrieved dSCDs for the measured show as well negative as





positive values. The differences of the results for synthetic and measured spectra indicate that the measurements are strongly influenced by factors like instrumental properties, which don't affect the synthetic spectra. For both data sets, the smallest deviations from zero are found for lower fit boundaries of 306 and 307 nm. The choice of the upper fit boundary has almost no influence on the derived $SO_2$ dSCDs. The much stronger influence of the choice of the lower fit boundary is expected, because

5  of several reasons:

- the $SO_2$ absorptions increase strongly towards short wavelengths

- the radiance decreases strongly towards short wavelengths

- the $O_3$ absorption strongly increases towards short wavelengths

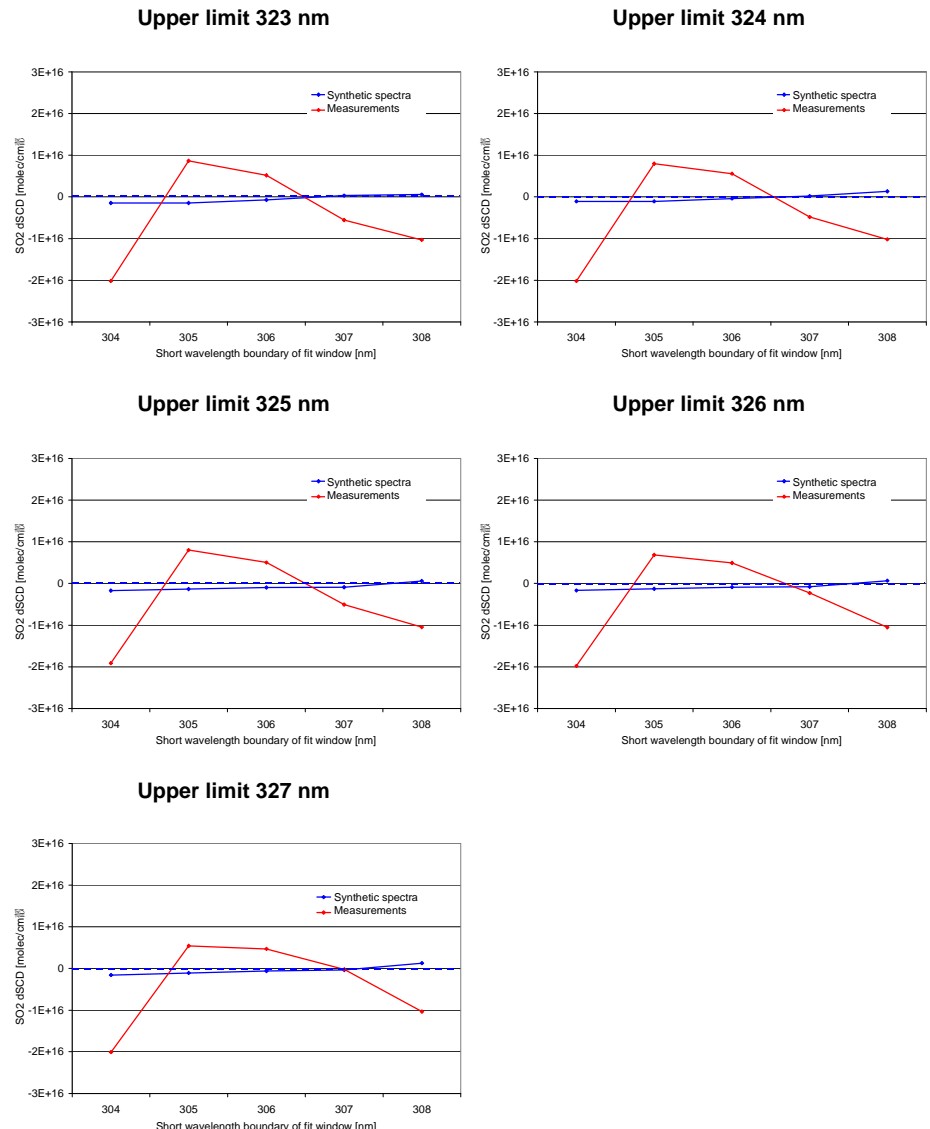

**Fig. A12** Dependence of the fit results (for 1° elevation) for the measured spectra (red lines) and synthetic spectra (blue lines) on the upper and lower boundaries of the fit range. Also shown are the 'true dSCDs' derived from the simulated air



mass factors (blue dashed lines). The individual plots represent results for one fixed upper wavelength limit. The lower wavelength limit is represented by the x-axes.

**Fit error and RMS**

5     Fig. A13 shows the $SO_2$ fit errors and RMS for synthetic (left) and measured spectra (right). Like for the analysis of BrO and HCHO, also for $SO_2$ systematically higher fit errors and RMS are found for the measured spectra. However, compared to BrO and HCHO, the difference is much larger for $SO_2$. This again indicates that the analysis of the measured spectra in the $SO_2$ analysis range is strongly affected by effects, which are not included in the simulation of the synthetic spectra (e.g. imperfect cross section, imperfect convolution). Thus for $SO_2$, only limited conclusions can be drawn from the synthetic spectra for the

10    analysis of the real spectra. From the results of the measured spectra, the best lower limit of the fit range is probably found at about 305 – 306 nm (fit error) and 308 (RMS). For the upper limit of the fit range no clear dependence is found.

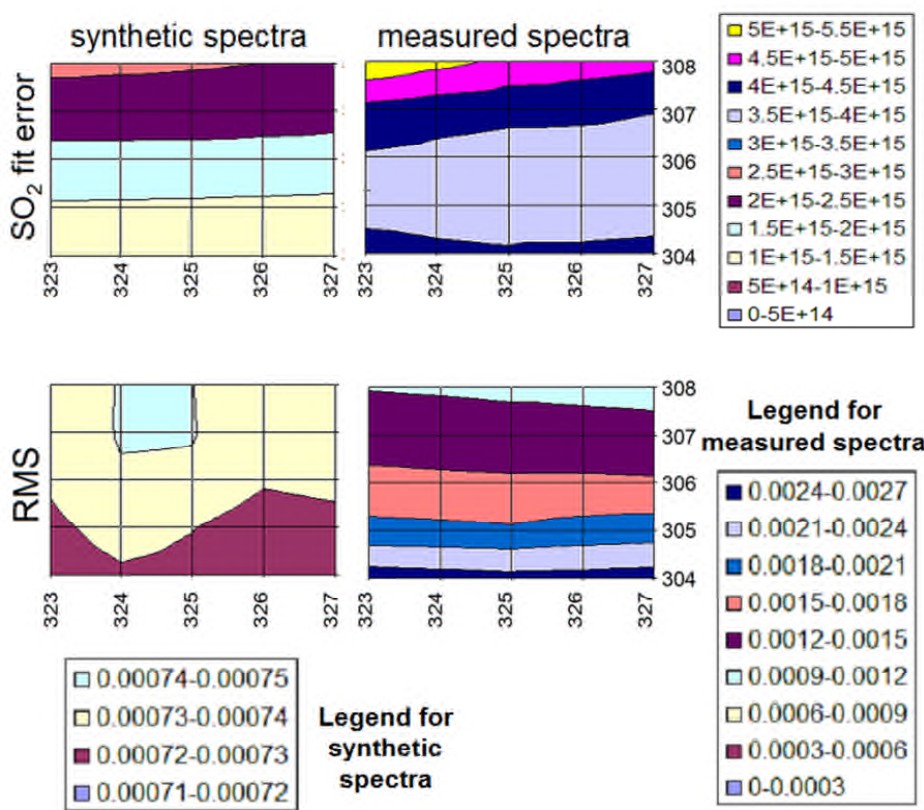

**Fig. A13** Dependence of the $SO_2$ fit errors and RMS on the upper and lower boundaries of the fit range for the synthetic (left) and measured spectra (right).





**Standard deviation of the retrieved dSCDs and correlation between the dSCDs of $SO_2$ and $O_3$**

In this section, the temporal variability of the derived $SO_2$ dSCDs and the correlation between the dSCDs of $SO_2$ and $O_3$ for $1°$ elevation are investigated. Although we don't make specific use of the $O_3$ retrieval results in this study, the correlation between the dSCDs of $SO_2$ and $O_3$ can be used to assess the quality of the $SO_2$ fit results. For the synthetic spectra the retrieved $SO_2$

dSCDs should be (close to) zero, because no $SO_2$ absorption was included in the simulation of the synthetic spectra. Also the $O_3$ absorption of all spectra should be the same.   Accordingly, the correlation between the dSCDs of $SO_2$ and $O_3$ should also be (close to) zero. Some temporal variation is of course expected because of the added noise. For the measurements, the atmospheric trace gas concentrations can vary, and also changes of the atmospheric visibility might further contribute to a variation of the measured trace gas dSCDs. Thus the retrieved variability of the trace gas dSCDs for the measured spectra is

expected to be higher than for the synthetic dSCDs. For the measured spectra, an (anti-) correlation of the derived dSCDs of $SO_2$ and $O_3$ might also reflect a (anti-) correlation of their true atmospheric absorptions. However, this is not very probably given the rather different profile shapes and formation processes. Thus a low correlation between the dSCDs of $SO_2$ is considered as an indication for a good fit quality also for the measured spectra.

Figure A14 shows the standard deviation of the time series of the $SO_2$ dSCDs for $1°$ elevation angle as well as the

correlation coefficient between the dSCDs of $SO_2$ and $O_3$ for the synthetic (left) and measured spectra (right). As expected the standard deviations are higher for the measured spectra. Also the correlation between both species is higher for the measured spectra. Like for the fit error and the RMS, the differences between the results for the synthetic and measured spectra are much higher than for the BrO and HCHO analysis. This again indicates that for $SO_2$, only limited conclusions can be drawn from the synthetic spectra for the analysis of the real spectra. From the results of the measured spectra, the best lower limit of the fit

range is found to be about $306 – 307$ nm ($SO_2$ std) and $304$ or $306$ (correlation). For the upper limit of the fit range no clear dependence is found.





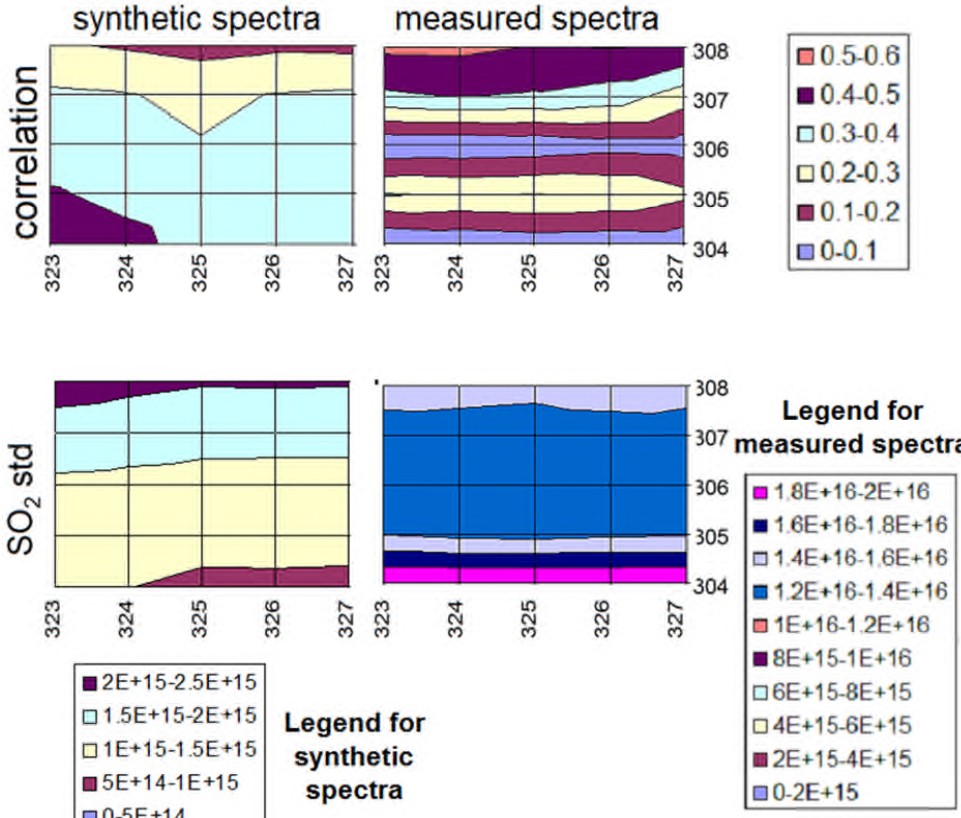

**Fig. A14** Dependence of the variation (expressed as standard deviation) of the derived $SO_2$ dSCDs for 1° elevation angle and the correlation between the $SO_2$ and $O_3$ dSCDs (expressed as r²) on the upper and lower boundaries of the fit range (left: synthetic spectra; right   measured spectra.

**Final choice of fit the range**

Based on the results from the previous sub section, the fit range from 306 to 325 nm is chosen for the analysis of $SO_2$. This choice is mainly based on the results for the measured spectra, because the synthetic spectra were found to be not

10    representative for the real measurements (see discussion above). For the chosen fit range the lowest values of the fit error, the standard deviation of the $SO_2$ dSCD and the correlation between the dSCDs of $SO_2$ and $O_3$ were found.

**Error estimate**

The systematic and random uncertainties of the fit results for individual spectra (average of 10 original spectra) are

15    summarized in Table A4.





**Table A4** overview on the different systematic and random error sources

| error source | error of SO$_2$ fit result |
|---|---|
| | |
| **systematic errors** | |
| Comparison of the fit results for the synthetic spectra to the 'true dSCDs' | 8e15 molec/cm² |
| | |
| **random errors** | |
| Fit error | 4e15 molec/cm² |
| Standard deviation* | 1.3e16 molec/cm² |

*the standard deviation describes the total statistical error; at least part of the total statistical error is caused by the fit error.

The systematic error of the dSCDs is usually dominated by systematic effects of the spectral retrieval. These systematic
5    effects are quantified by the comparison of the fit results for the measured and synthetic spectra. Only for high trace gas dSCDs
the uncertainty of the cross section might become important. But for the rather low SO$_2$ dSCDs retrieved in this study the
uncertainty of the cross section can be neglected.

For individual measurements, the random errors dominate the total uncertainty. However, if several measurements are
averaged these errors can be largely reduced.

10    **Quality filter**

Spectra with bad fit quality have to be removed from further processing. First, measurements with a small number of individual
scans (<800) are removed. This filter removes the worst fit results. In addition, a RMS filter is applied. Here a threshold for the
RMS of 1.8e-3 is chosen. Figure A15 shows the frequency distribution of the RMS (for measurements with more than 800
scans). The RMS threshold removes about 24% of the SO$_2$ results.



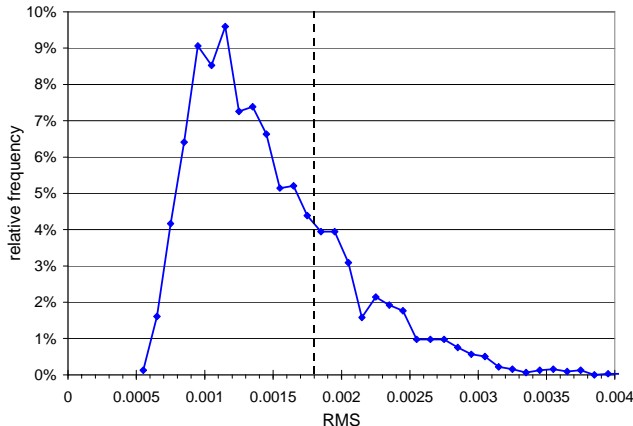

**Fig. A15** Frequency distribution of the RMS of the spectral analysis of $SO_2$ for clear sky conditions. The vertical line indicates the threshold (0.0018) for removing measurements with high fit errors.

5    **A4.3 Determination of the optimum fit range for $NO_2$**

For the $NO_2$ analysis the exact determination of the fit range is less critical than for BrO, HCHO, and $SO_2$, because of less atmospheric interfering. Moreover, the signal to noise ratio in the blue spectral range is larger than in the UV, and in particular it is more constant over the entire $NO_2$ fit range. Because of these reasons, less effort is spent on the determination of the $NO_2$ fit range, and in particular no synthetic spectra are used.

10    The spectral range of the instrument is 290 to 437 nm. Thus in principle a rather large spectral range from about 400 nm to 436 nm could be used. However, it turned out that large fit residuals occur for wavelength ranges with upper boundaries > 427nm (and also below 399nm, see Fig. A16). Thus for the $NO_2$ analysis a fit range between about 399 and 427 nm was chosen.





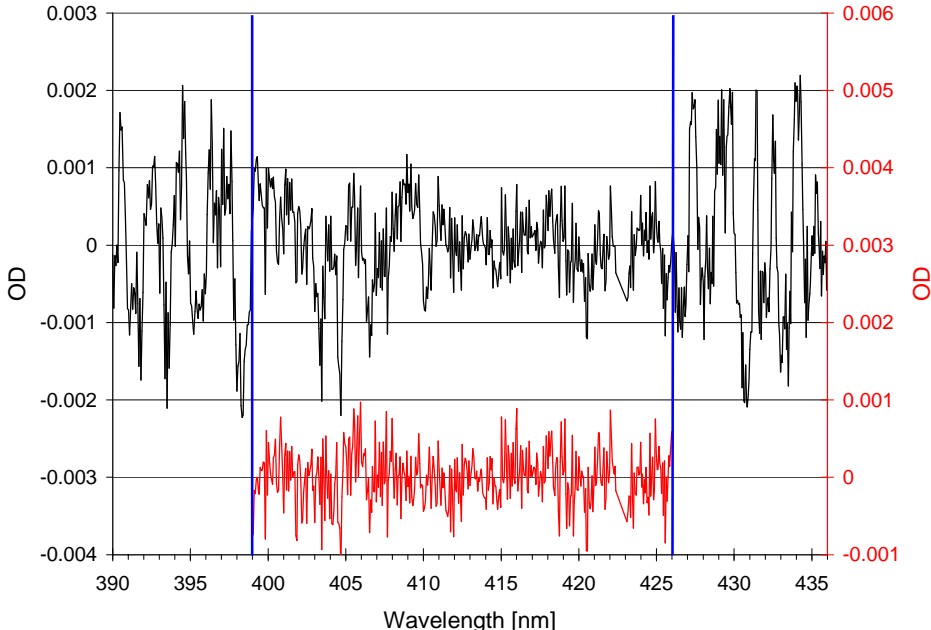

**Fig. A16** Fit residuals for a NO$_2$ fit in two wavelength ranges (black: 390 – 436nm, left y-axis; red: 399 – 426 nm, right y-axis). The measurement (average of 10 individual measurements) was taken on 2 December 2012, 04:51 to 7:00. The (average) SZA and elevation angle were 59.3° and 6°, respectively. The residual of the large spectral range shows strong systematic structures below 400nm and above 427 nm (RMS: 7.8e-4). The residual of the small spectral range shows much less systematic structures (RMS: 3.5e-4). The blue lines indicate the lower and upper boundaries of the finally chosen fit range 399 – 426nm.

To further specify the lower and upper fit boundaries, the RMS, fit errors, and the standard deviation for 1° elevation angle were derived for clear sky observations. The dependencies of these quantities on variations of the upper and lower fit boundaries are shown in Fig. A17. As expected, only weak dependencies were found. The final choice of the fit range was 399 – 426, because for that range the fit error and the standard deviation for measurements at 1° elevation were found to be smallest. However, the results for the other investigated fit ranges are very similar (the deviations are between -2e14 and +4e14 molec/cm², see Fig. A17.



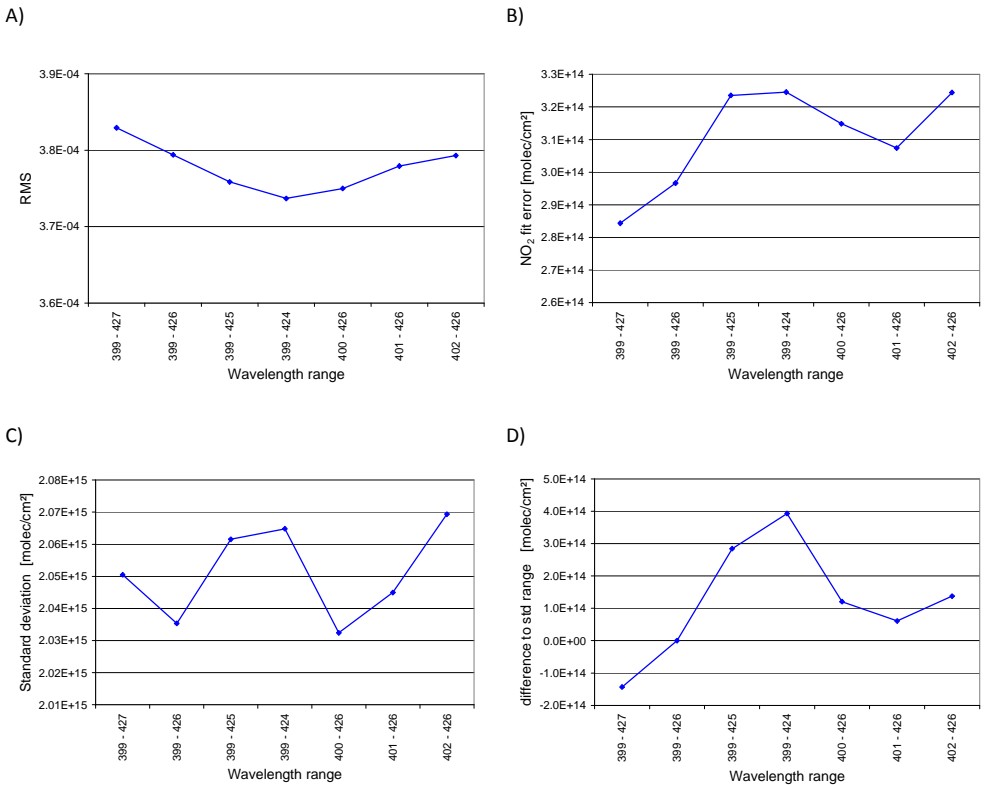

**Fig. A17** Comparison of the fit results for different $NO_2$ fit ranges. A) Mean RMS; B) Mean $NO_2$ fit error; C) Standard deviation of $NO_2$ dSCDs for 1° elevation angle; D) Mean difference of the $NO_2$ dSCDs with respect to the results for the fit range 399 – 426 nm.

**Error estimate**

The systematic and random uncertainties of the fit results for individual spectra (average of 10 original spectra) are summarized in Table A5.

10    **Table A5** overview on the different systematic and random error sources

| error source | error of $NO_2$ fit result |
|---|---|
|  |  |
| systematic errors |  |





| Deviation of NO$_2$ dSCDs for different fit ranges | 6e14 molec/cm² |
|---|---|
| | |
| **random errors** | |
| Fit error | 3e14 molec/cm² |
| Standard deviation* | 2e15 molec/cm² |

*the standard deviation describes the total statistical error; at least part of the total statistical error is caused by the fit error.

The systematic error of the dSCDs is usually dominated by systematic effects of the spectral retrieval. These systematic effects are quantified by the comparison of the fit results for the different spectral ranges. Only for high trace gas dSCDs the uncertainty of the cross section might become important. But for the rather low NO$_2$ dSCDs retrieved in this study the uncertainty of the cross section can be neglected.

For individual measurements, the random errors dominate the total uncertainty. However, if several measurements are averaged these errors can be largely reduced.

**Quality filter**

Spectra with bad fit quality have to be removed from further processing. First, measurements with a small number of individual scans (<800) are removed. This filter removes the worst fit results. In addition, a RMS filter is applied. Here a threshold for the RMS of 5e-4 is chosen. Figure A18 shows the frequency distribution of the RMS (for measurements with more than 800 scans). The RMS threshold removes about 10% of the NO$_2$ results.

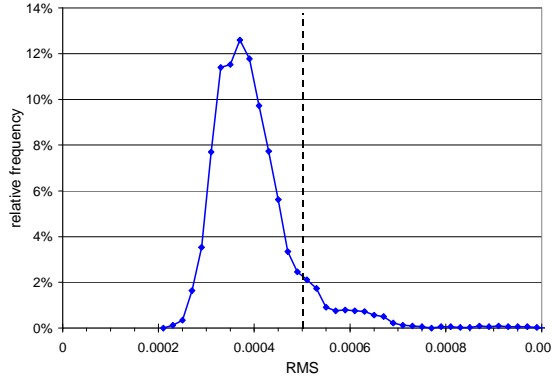

**Fig. A18** Frequency distribution of the RMS of the spectral analysis of NO$_2$ for clear sky conditions. The vertical line indicates the threshold (0.0005) for removing measurements with high fit errors.





**A5 Influence of stratospheric trace gas abundances on the MAX-DOAS results**

**A5.1 NO$_2$**

In this section the influence of the stratospheric NO$_2$ absorption on the MAX-DOAS measurements at the WLG station are investigated. The underlying question is whether the simulation of the stratospheric and tropospheric absorptions can be separated (which makes the direct interpretation of the MAX-DOAS results much easier). To answer that question, the NO$_2$ dSCDs observed by the MAX-DOAS measurements corresponding to the stratospheric NO$_2$ absorption are simulated. For the

radiative transfer simulations stratospheric NO$_2$ profiles provided by the study from Bauer et al. (2012) were used. They provide profiles for different seasons and latitude bands. Since the WLG station is located close to the border between two latitude bands (30°S to 30°N, and 30° to 60°N), the average of the profiles of both latitude bands were used. Two stratospheric profiles were derived, one for summer and one for winter (see Fig. A19).

    In Fig. 20 the corresponding dSCDs simulated for the MAX-DOAS measurements at different elevation angles are shown.

The deviations of the NO$_2$ dSCDs from zero are largest for 1° elevation angles. They are about -1e14 molec/cm in winter and between -2e14 molec/cm and -1e14 molec/cm in summer. Thus they are about one order of magnitude smaller than the observed NO$_2$ dSCDs (see Sect. 7). For the quantitative interpretation of the measured NO$_2$ dSCDs they can therefore be neglected.

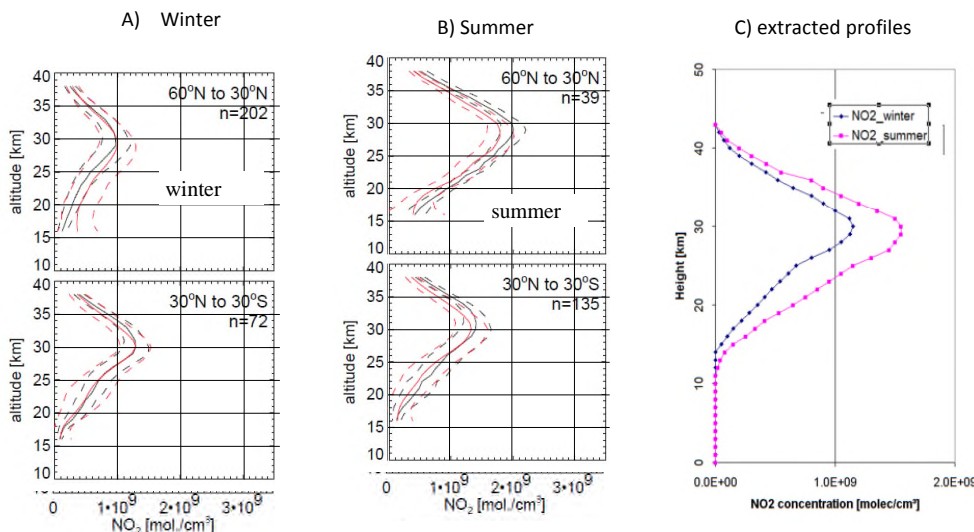

**Fig. A19** Stratospheric profiles taken from the study of Bauer et al. (2012) for Winter (A) and Summer (B) for the tropics and northern mid-latitudes. The right figure shows the extracted profiles for summer and winter, which were used for the radiative transfer simulations. The corresponding VCDs are: 1.51e15 (winter) and 2.30e15 (summer).





Winter                                    Summer

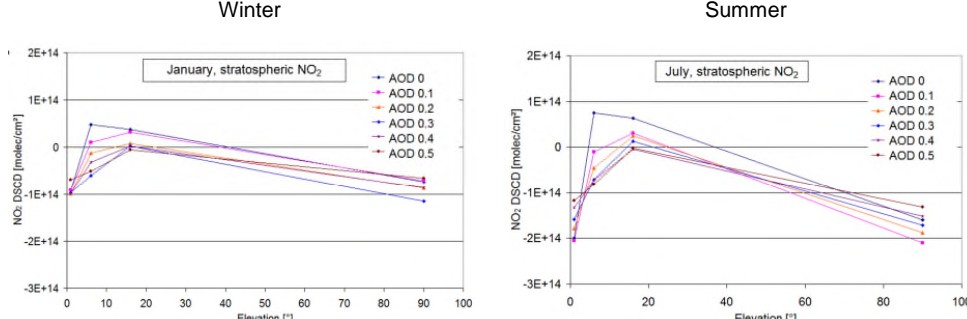

**Fig. A20** Simulated NO$_2$ dSCDs for the stratospheric profiles shown in Fig. A19 (right) for winter (left) and summer (right).

### A5.2 BrO

The stratospheric BrO profiles were constructed following the study of Dorf et al. (2008) who described the stratospheric BrO mixing ratio relative to the altitude of the tropopause height (Fig. A21 left). We used their parameterisation, but with a slightly

10  lower maximum BrO mixing ratio of 15 ppt instead of 16 ppt in order to account for the decrease of the stratospheric BrO load between the study of Dorf et al. (2008) and the measurements considered in this study. The two sub figures at the right side of Fig. A21 show the derived BrO height profiles for different tropopause layer heights. In part B) of Fig. A21 it is assumed that no BrO exists in the troposphere; in part C) a background BrO mixing of 1 ppt was assumed. The corresponding BrO dSCds are shown in Figs A22 and A23. Interestingly, negative BrO dSCDs are found for measurements at 1° elevation angle if no BrO

15  was assumed in the troposphere (Fig. A22). Substantially higher BrO dSCDs are found for the cases when a tropospheric background concentration of 1 ppt was assumed (Fig. A23).





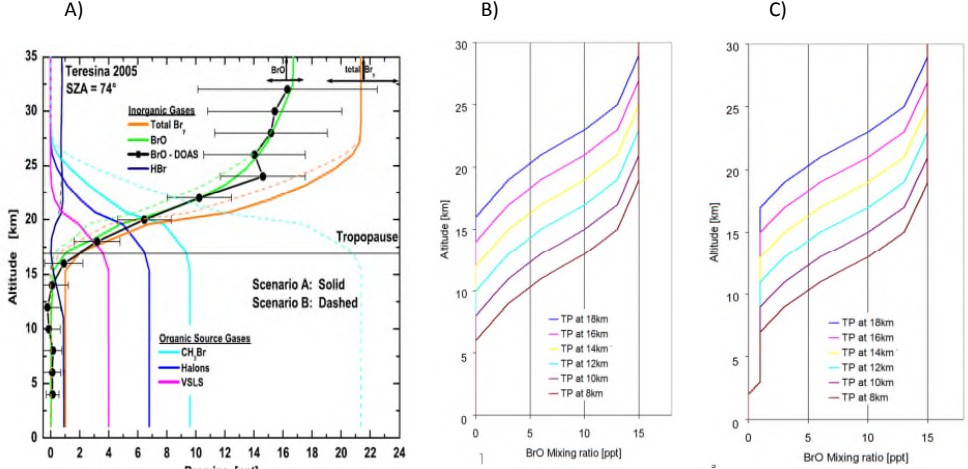

**Fig. A21** Left: as Fig. 1 in Dorf et al. (2008): The green line represents the BrO mixing ratios derived from various measurements and model simulations as function of the relative height with respect to the tropopause (horizontal black line). In the two figures at the right the corresponding BrO mixing ratios are plotted as function of different tropopause heights. In Fig. B) no BrO was assumed in the troposphere; in Fig. C) a constant BrO mixing ratio of 1 ppt was assumed in the troposphere. Note that the decrease of the stratospheric BrO mixing ratio between 2008 and the period of the Tibet measurements of about -1 ppt was taken into account.

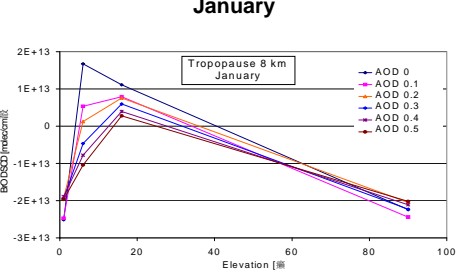

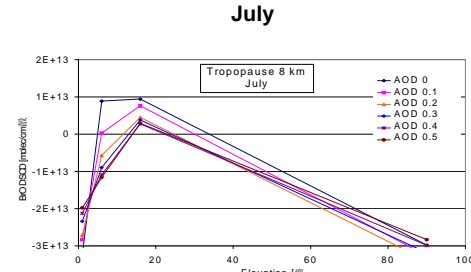





**Fig. A22** BrO dSCDs calculated for the profiles shown in Fig. A21B with zero BrO concentrations in the troposphere for different tropopause heights (left: winter; right: summer).



**Fig. A23** BrO dSCDs calculated for the profiles shown in Fig. A21C with 1 ppt BrO in the troposphere for different tropopause heights (left: winter; right: summer).





**A6 Statistics of the different sky conditions for the individual trace gas results**

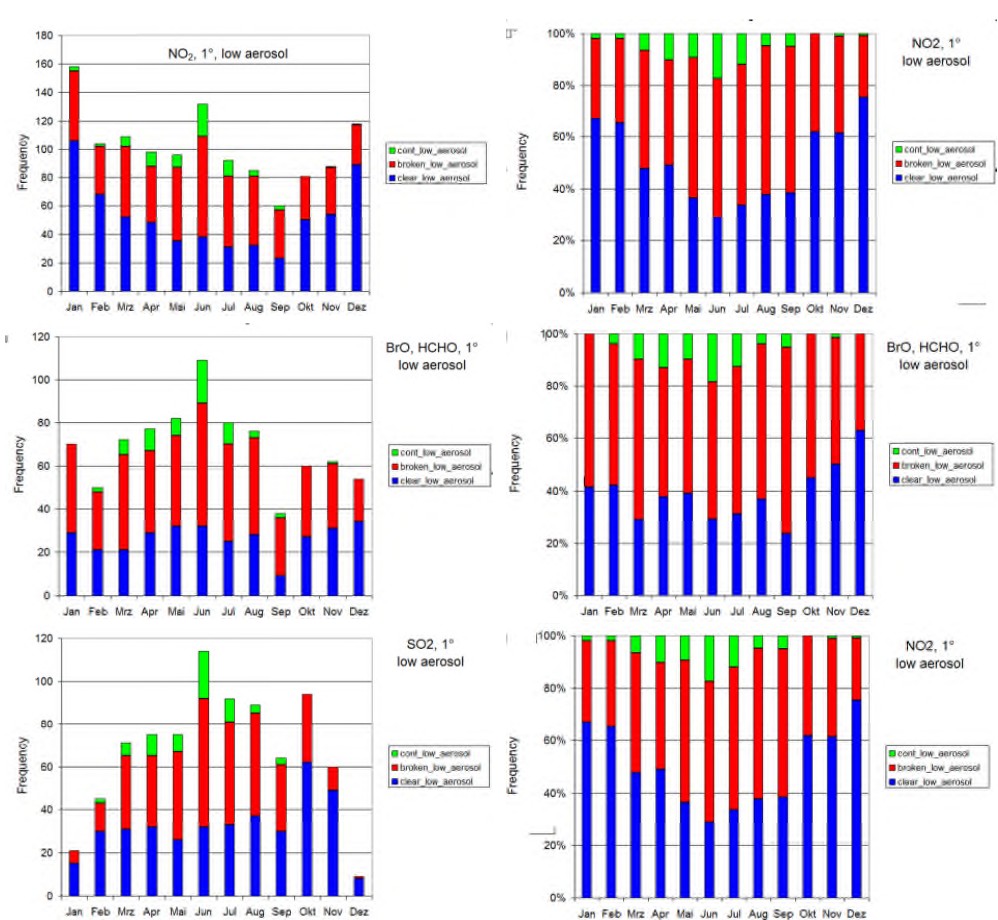

**Fig. A24** Absolute (left) and relative (right) frequency of the different sky conditions for results of selected trace gases (top: NO$_2$, middle: BrO and HCHO, bottom: SO$_2$). The statistics are based on the number of observations at 1° elevation angle (mean of 10 original spectra from April 2012 to April 2015). In addition to the filter for the removal of high aerosol loads, also the specific RMS filters for the different trace gases are applied (see appendix A4).



**A7 Dependencies of the trace gas results on wind speed and direction**

Dependence on wind speed

| NO₂ whole year | NO₂ different seasons | SO₂ whole year | SO₂ different seasons |
|---|---|---|---|

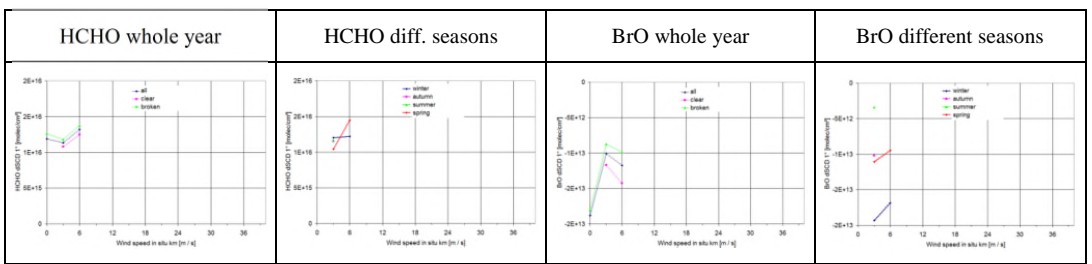



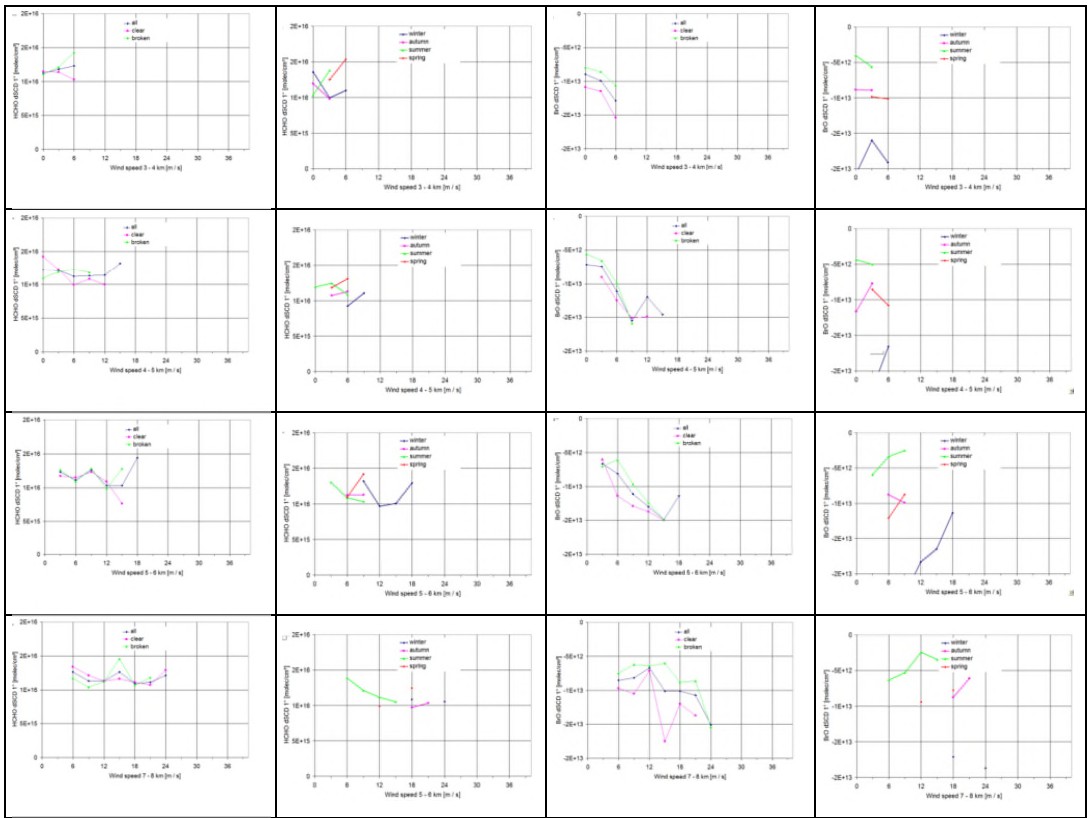

**Fig. A25** Dependencies of the trace gas dSCDs measured at 1° elevation on the wind speed for the altitude layers: 3-4km; 4-5km; 5-6km; 7-9km. These layers were selected because they represent the overall variability of wind speed and direction between 3 and 10 km. For the wind speed bins of 3 km/s were chosen. Only data with more than 20 data points per bin are shown. Dependencies for the whole year (and different cloud conditions) as well as for different seasons (for combined measurements for clear sky and broken clouds) are shown (seasons: winter: DJF, spring: MAM, summer: JJA, autumn: SON).

Dependence on wind direction

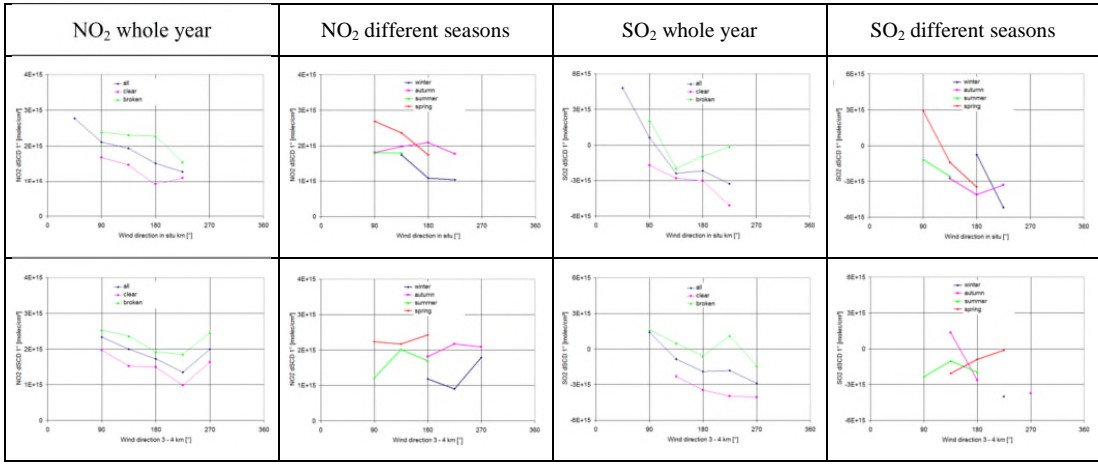



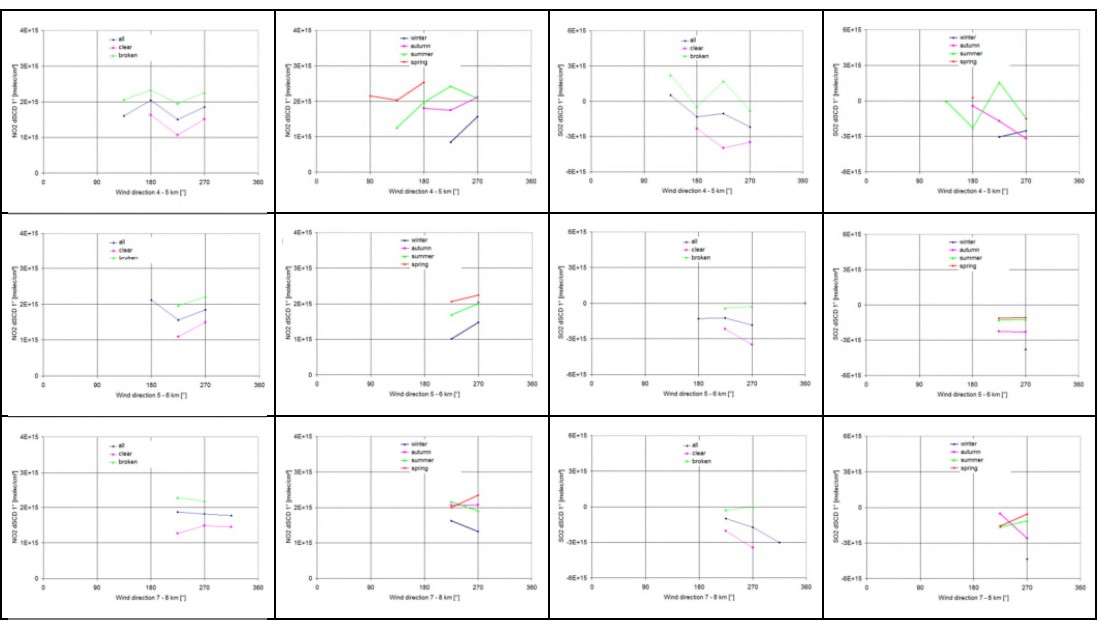

Dependence on wind direction

| HCHO whole year | HCHO diff. seasons | BrO whole year | BrO different seasons |
|---|---|---|---|

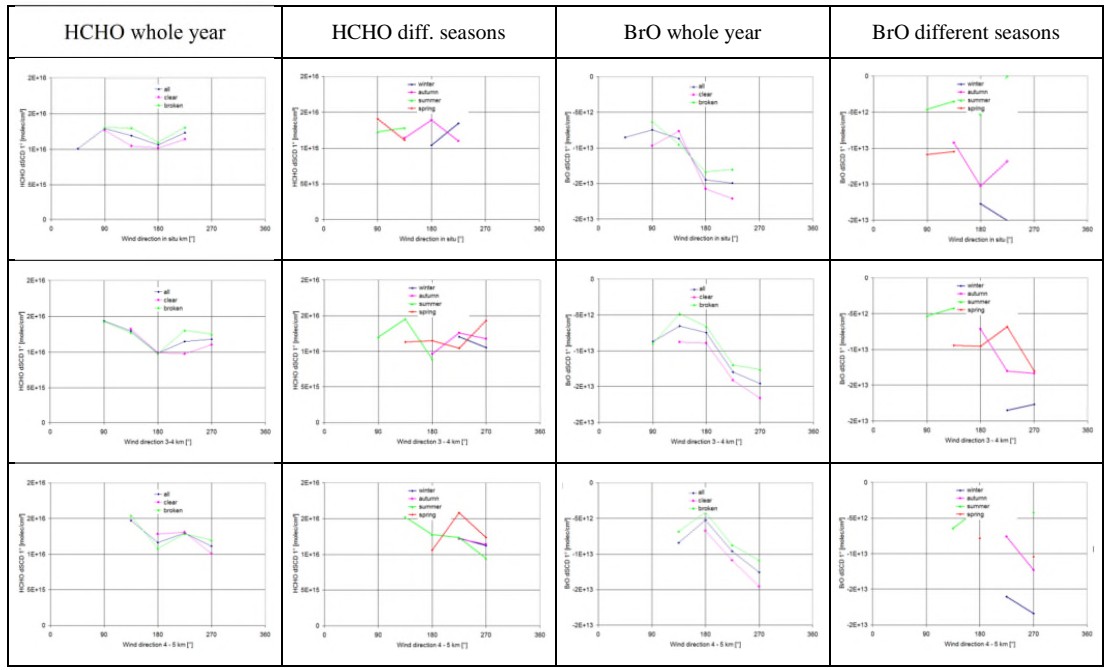



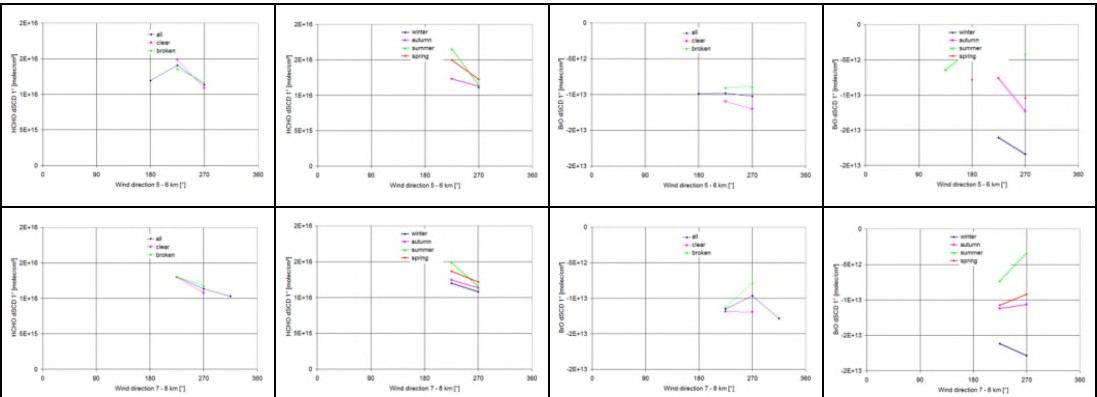

**Fig. A26** Dependencies of the trace gas dSCDs measured at 1° elevation on the wind direction for the altitude layers: 3-4km; 4-5km; 5-6km; 7-9km. These layers were selected because they represent the overall variability of wind speed and direction between 3 and 10 km. For the wind direction bins of 45° were chosen. Only data with more than 20 data points per bin are shown. Dependencies for the whole year (and different cloud conditions) as well as for different seasons (for combined measurements for clear sky and broken clouds) are shown (seasons: winter: DJF, spring: MAM, summer: JJA, autumn: SON). Note that the wind directions are almost entirely between 90° and 300°. Thus no potential ambiguity for wind directions around 0°/360° affects these results.

**Data availability**

The spectral analysis and RTM simulation results here are available upon request.

**Author contributions**

JM and TW designed the study. JM, JJ, JG, ZZ, JW, PL and GZ contributed to the measurements. TW, JM, SDörner, SDonner, JJ, SC, JP and JL contributed to the data analyses. TW and JM prepared the manuscript with consented by all co-authors.

**Competing interests**

The authors declare that they have no conflict of interest.

**Special issue statement**

This article is part of the special issue "Study of ozone, aerosols and radiation over the Tibetan Plateau (SOAR-TP) (ACP/AMT inter-journal SI)". It is not associated with a conference.

**Acknowledgements.**

This work was funded by MOST (grant no. 2018YFC1505703) and NSFC (grant nos. 41275140 and 41875146) and CMA (grant no. GYHY201106023). We thank all the staff from WLG for assisting measurement work.



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
