# Peer review of "MAX-DOAS measurements of NO2, SO2, HCHO and BrO at the Mt. Waliguan WMO/GAW global baseline station in the Tibetan Plateau"

_Atmospheric Chemistry and Physics, 2019_

## Referee Comment (RC1) · Anonymous Referee #1 · 12 Feb 2020

Review of Âń MAX-DOAS measurements of NO2, SO2n HCHO and BrO at Mt Wliguan WMO/GAW global baseline station in the Tibetan Plateau Âż by Jianzhong Ma et al.

The objective of the paper is to describe the measurements and the results of five years (2012-2015) monitoring of four atmospheric chemical species (NO2, SO2, HCHO and BrO) at a high-altitude station on the Tibetan plateau by the MAX-DOAS technique. It is shown that the averaging of 10 spectra allows to improve the signal to noise ratio, allowing to measure NO2 varying from 5 ppt in winter and 70 ppt in summer in the lower layer, whereas the HCHO is of the order of 0.3-0.7 ppb and that of SO2 is lower than 0.5 ppb and that of BrO not significant.

[Figure]

General comments The paper is very long which includes 23 pages with 18 complex figures and another 28 pages and 26 figures in appendix. Moreover, the figures are often repeating and include up to 12 plots. In addition there are 103 references often unnecessary. The conclusions, as summarised, above are very limited. In summary the paper is almost impossible to follow.

Recommendation There is no doubt that the paper requires major revision before acceptance, including: - simplified organisation following the list of important items, (and ignoring less important), - list of figures really required (e.g wind direction and speed, tropopause height, correlations)? - Figure 8, 9, 10, 11b.. ? A1, A2(already in main text), A9, A10, .. and many others disputable - Correction of figures difficult to read (e.g. 1b and too small legends) - Order of species in the discussion: NO2, HCHO, SO2, BrO, identical in the various tables and figures, - Why broken cloud periods shown? Is fig A24 necessary? Why not using clear sky only? - Increase figures legends e.g. A2 right, A3right. . ..and many others - Figures dates: what Jan 1, Jan 13 . . . mean? (e.g. A2, A3 . . .) - Why correlation coefficient between meteorological quantities?

---

## Referee Comment (RC2) · Anonymous Referee #2 · 28 Feb 2020

In their manuscript "MAX-DOAS measurements of NO2, SO2, HCHO and BrO at the Mt. Waliguan WMO/GAW global baseline station in the Tibetan Plateau", the authors report on three years of MAX-DOAS measurements at the high altitude station of Waliguan. They provide a detailed description of the technical steps taken in the evaluation of the spectra, describe the radiative transfer calculations performed and discuss connections between the measurement results and meteorological parameters.

Measurements of high altitude background concentrations of atmospheric trace gases and aerosols are an interesting topic, and MAX-DOAS measurements are one sensitive measurement technique to obtain long-term data sets of such quantities.

[Figure]

Unfortunately, this manuscript in mainly a long and very detailed description of the technical aspects of the analysis and the meteorology at the station and provides very little results which are of general interest. I can therefore not recommend this manuscript for publication. I'd suggest that the authors work on the manuscript by removing all unnecessary parts, tightening the technical discussions to what is really needed, and focus more on the results and what we can learn from them. Such a shortened and more focused study could then be re-submitted to ACP or another journal.

**Major comments**

- The manuscript introduces and discusses many meteorological parameters and their correlations with the measurements, but all correlations are very small and this whole part could be summarised in a single sentence: No significant correlation was found between any of the measured quantities and meteorological parameters. Removing all the unnecessary figures and descriptions on meteorology would already considerably reduce the length of the manuscript.

- Most of the manuscript deals with technical aspects of the retrieval and in particular in the supplemental material, sensitivity studies are described in great detail. While thorough documentation of the methods used is a good thing, most of what is described is state of the art and could have been summarised in a few short sections.

- The one point where the authors introduce a better approach than earlier studies, namely accounting for the topography along the line of sight is unfortunately not expanded upon at all – there is no discussion of what the difference to a standard analysis using a representative surface altitude and a 1d retrieval would have been, how these 2d effects vary with season, snow cover and aerosol optical depth and if this approach results in more accurate results considering that due to computational time limitations, only two scenarios could be computed.

- Unfortunately, the measurements were performed with a simple, not very sensitive instrument and are therefore not of high quality. Because the shading tube was missing, zenith measurements could not be used which leads to a further reduction in sensitivity. The fact that elevation calibration appears to be off by 4° further increases uncertainties, in particular as this implies that the authors cannot be sure about the exact pointing of the horizontal measurements they use. This uncertainty needs to be evaluated and included in the error budget.

- There is literally no discussion of the results in terms of comparison to other measurements or model results or what we can learn from the three years of measurements which are presented.

**Minor comments**

- I believe that the excessive listing of references at the top of page 4 makes no sense as these references have no connection to the topic of this manuscript.

- I'm not so sure about the discussion of straylight on page 8. My guess is, that the straylight correction does not primarily correct for more straylight in the reddish direct sun spectra (cloudy sky spectra should have a similar wavelength distribution as direct sun spectra) but that it corrects for the change in wavelength dependence of Fraunhofer filling in.

- It should at least be mentioned that zenith measurements which are used here for the CI were earlier discarded for trace gas retrievals because of suspected direct sun impact

- Figure 8: Are these realistic stratospheric BrO profiles? This would imply a factor of 4 variability in stratospheric BrO columns – has this been observed in measurements?

- Figure 10: Title of figures not correct

- Figure 11: t does not become clear if all directions are included in the statistics and if so why. My understanding is that only 1° measurements were used in a quantitative way.

- Conclusions, line 22: Enhanced trace gas mixing ratios – enhanced in comparison to what?

- I'm surprised that no reference is made to the paper

  Gomez, L., Navarro-Comas, M., Puentedura, O., Gonzalez, Y., Cuevas, E. and Gil-Ojeda, M.: Long-path averaged mixing ratios of O3 and NO2 in the free troposphere from mountain MAX-DOAS, Atmos. Meas. Tech., 7(10), 3373–3386, doi:10.5194/amt-7-3373-2014, 2014.

  which uses a similar method for a related topic.

---

## Author Comment (AC1) · 1 Apr 2020

Referee comments are in black. Author responses are in blue.

**General comments.** The paper is very long which includes 23 pages with 18 complex figures and another 28 pages and 26 figures in appendix. Moreover, the figures are often repeating and include up to 12 plots. In addition there are 103 references often unnecessary. The conclusions, as summarised, above are very limited. In summary the paper is almost impossible to follow.

We thank the anonymous referee for his/her insightful comments and constructive suggestions, which are very helpful for improving our manuscript. We made major revisions accordingly. The Appendix and other parts that introduce detailed technical methods, settings and related parameters have been moved to the Supplement. Some references (e.g., literatures about previous measurement work at WLG) and figures (e.g., wind speed/direction and the effects on the trace gas results) are skipped over. We have reorganized the structure of the paper so that it is easy to follow on our best. Below enclosed is the revised version of the paper.

**Recommendation.** There is no doubt that the paper requires major revision before acceptance, including:
- simplified organisation following the list of important items, (and ignoring less important),
  The paper has been reorganized, and the items that the wider field readers may not care about have been moved to the Supplement.

- list of figures really required (e.g wind direction and speed, tropopause height, correlations)?
  Figures about correlations with meteorological parameters have been removed.

- Figure 8, 9, 10, 11b.. ? A1, A2 (already in main text), A9, A10, . and many others disputable
  Figures 8, 9, 10 and all figures in the Appendix have been moved to the Supplement or skipped over.

- Correction of figures difficult to read (e.g. 1b and too small legends)
  Corrected.

- Order of species in the discussion: NO$_2$, HCHO, SO$_2$, BrO, identical in the various tables and figures,
  Done.

- Why broken cloud periods shown? Is fig A24 necessary? Why not using clear sky only?
  Figure A24 has been moved to the Supplement (Fig. S28). The right part of old Fig. 13, which is for broken clouds, has been moved to the Supplement (Fig. S29). Only the results for the clear sky are presented in the main body of the revised manuscript (Fig. 8).

- Increase figures legends e.g. A2 right, A3right: : :.and many others
  Done.

- Figures dates: what Jan 12, Jan 13 : : : mean? (e.g. A2, A3 : : :)
  They mean 1 Jan 2012, 1 Jan 2013 and so on, which have been added to the figure legends.

- Why correlation coefficient between meteorological quantities?
  This part has been removed.

[revised manuscript text omitted]

Average O$_4$ dSCDs for clear sky conditions

[Figure]

Average O$_4$ dSCDs for broken clouds

[Figure]

**Fig. S27** O$_4$ dSCDs at 1 ° elevation for clear sky **(top)** and broken clouds **(bottom)** spectra (averages of 10 original spectra for 2012 - 2015) with number of scans > 800 and RMS of the O$_4$ fit < 2e-3.

[Figure]

**Fig. S28** Absolute **(left)** and relative **(right)** frequency of the different sky conditions for results of selected trace gases (top: $NO_2$, middle: BrO and HCHO, bottom: $SO_2$). The statistics are based on the number of observations at 1° elevation angle (mean of 10 original spectra from April 2012 to April 2015). In addition to the filter for the removal of high aerosol loads, also the specific RMS filters for the different trace gases are applied (see Supplement Sect. 3.3).

**7 Seasonal means of the dSCDs under broken cloud conditions**

[Figure]

**Figure S29** Seasonal means of the trace gas dSCDs for different elevation angles for broken clouds and low aerosol load. For $NO_2$, $SO_2$, and HCHO the right axes represent the approximate mixing ratios for measurements at 1 ° elevation angle. The blue dotted lines indicate the systematic uncertainties, which can be considered as lower bound of the detection limit.

**8 Estimation of tropospheric BrO mixing ratios from the measured BrO dSCDs**

Fig. S30 shows the dependence of the measured BrO dSCDs for different elevation angles together with simulation results for tropospheric background mixing ratios of 0 ppt or 1 ppt. While for 1 °elevation a strong dependence on the tropopause height is found, such a dependency is not observed for the higher elevation angles. These findings are consistently found for the measurements and simulations (but the measured BrO dSCDs show a rather large scatter). Interestingly, the differences between the simulated BrO dSCDs for 0 and 1 ppt tropospheric background are almost constant. This finding indicates that the increase of the measured BrO dSCDs (with respect to a simulation without a tropospheric BrO background) caused by an increase of the tropospheric BrO background is almost proportional to the tropospheric BrO mixing ratio.

For the estimation of the (upper limit of the) tropospheric BrO background the measurements at 1 °elevation angle are chosen, because they are most sensitive to BrO absorptions in the troposphere (the difference between the simulations for 0 and 1 ppt in Fig. S30 are largest for an elevation angle of 1 °). To account for the dependence on the tropopause height the following procedure was applied:

First the measured BrO dSCDs are subtracted from the simulated BrO dSCDs for the same tropopause height. This is done for the simulations for 0 ppt as well as 1 ppt background BrO. From the obtained differences the man values and the standard deviations are calculated. For 0 ppt the mean difference is $0.9*10^{13}$ molec/cm²; for 1 ppt it is $2.4*10^{13}$ molec/cm². The standard deviation for both differences is $1.0*10^{13}$ molec/cm².

In the next step the total error (of the difference between simulations and measurements) is calculated as the sum of the standard deviation and the estimate for the systematic error of $0.3*10^{13}$ molec/cm² (see Supplement Sect. 3.3.3).

In the final step the total error ($1.3*10^{13}$ molec/cm²) is compared to the mean values of the differences for 0 ppt and 1 ppt. The total error is found above the mean difference for the simulations for 0 ppt BrO ($0.9*10^{13}$ molec/cm²), and below the mean difference for the simulations for 1 ppt BrO ($2.4*10^{13}$ molec/cm²). Assuming that the (increase of the) measured BrO dSCD depends linearly on the BrO background mixing ratio (see above), an upper limit for the BrO background mixing ratio of 0.23 ppt is obtained. From similar calculations for 6 ° and 16 ° elevation, upper limits of 0.34 ppt and 0.60 ppt are found, respectively.

[Figure]

**Fig. S30** Comparison of measured and simulated BrO dSCDs as function of the tropopause height for clear sky conditions. The blue and magenta lines represent simulation results for 1 ppt and 0 ppt BrO in the troposphere, respectively.

---

## Author Comment (AC2) · 1 Apr 2020

Referee comments are in black. Author responses are in blue.

**General comments.**
Measurements of high altitude background concentrations of atmospheric trace gases and aerosols are an interesting topic, and MAX-DOAS measurements are one sensitive measurement technique to obtain long-term data sets of such quantities. ACPD Unfortunately, this manuscript in mainly a long and very detailed description of the technical aspects of the analysis and the meteorology at the station and provides very little results which are of general interest. I can therefore not recommend this manuscript for publication. I'd suggest that the authors work on the manuscript by removing all unnecessary parts, tightening the technical discussions to what is really needed, and focus more on the results and what we can learn from them. Such a shortened and more focused study could then be re-submitted to ACP or another journal.

We thank the anonymous referee for his/her insightful comments and constructive suggestions, which are very helpful for improving our manuscript. We have shortened the manuscript by moving the Appendix and other parts, that provide very detailed descriptions of the technical aspects of the analysis and the meteorology, to the Supplement. Some references (e.g., literatures about previous measurement work at WLG) and figures (e.g., wind speed/direction and the effects on the trace gas results) are removed. We have reorganized the structure of the paper so that it is easy to follow by the readers in wider fields, not limited to experts in remote sensing. Comparisons of the results with previous studies on the levels of NO$_2$, SO$_2$, HCHO, and BrO in the remote low troposphere are added. We believe that measurements of high altitude background concentrations of these reactive trace gases are of general interest for the atmospheric chemistry community, and thus would like to submit the revised version of the manuscript (enclosed) for further review.

**Major comments**

- The manuscript introduces and discusses many meteorological parameters and their correlations with the measurements, but all correlations are very small and this whole part could be summarised in a single sentence: No significant correlation was found between any of the measured quantities and meteorological parameters. Removing all the unnecessary figures and descriptions on meteorology would already considerably reduce the length of the manuscript.
  Removed.

- Most of the manuscript deals with technical aspects of the retrieval and in particular in the supplemental material, sensitivity studies are described in great detail. While thorough documentation of the methods used is a good thing, most of what is described is state of the art and could have been summarised in a few short sections.
  Agreed. We have reorganized the manuscript, with the main results being kept in the main body of the manuscript and details being moved to the Supplement.

- The one point where the authors introduce a better approach than earlier studies, namely accounting for the topography along the line of sight is unfortunately not expanded upon at all

– there is no discussion of what the difference to a standard analysis using a representative surface altitude and a 1d retrieval would have been, how these 2d effects vary with season, snow cover and aerosol optical depth and if this approach results in more accurate results considering that due to computational time limitations, only two scenarios could be computed.

We performed additional radiative transfer simulations with flat surfaces, either at sea level or at 3700m altitude. It was found that the results for the elevated surface at 3700m are very similar to those using a realistic topography. In contrast the results for the flat surface at sea level differ strongly. These findings and the corresponding results are added to the revised version of the manuscript (in section 4). There also the following information about the influence of the surface albedo was added:

Simulations with high surface albedo representative for snow surfaces yielded almost the same results. This finding can be understood by the fact that the measurements and the Fraunhofer reference spectra are affected by changes of the surface albedo in the same way. Thus, for the retrieved dSCDs the effect of the surface albedo cancels out.

- Unfortunately, the measurements were performed with a simple, not very sensitive instrument and are therefore not of high quality. Because the shading tube was missing, zenith measurements could not be used which leads to a further reduction in sensitivity. The fact that elevation calibration appears to be off by 4° further increases uncertainties, in particular as this implies that the authors cannot be sure about the exact pointing of the horizontal measurements they use. This uncertainty needs to be evaluated and included in the error budget.

We admit that Mini MAX-DOAS is a simple instrument, but it is stable, solid, and was proven to be very suitable for long-term measurements under harsh environments like over the plateau. Concerning the elevation calibration, we disagree with the reviewer: From the comparison of the measurement and model results shown in Fig. S3 it can be concluded that the corrected elevation calibration is rather accurate. Deviations larger of 0.5 ° would be clearly obvious. Thus we conclude that the updated elevation calibration is correct within +/- 0.5 °

- There is literally no discussion of the results in terms of comparison to other measurements or model results or what we can learn from the three years of measurements which are presented.

As suggested, we have added a subsection (Sect. 6.3 in the revised version) to compare our results with previous studies, including measurements of $NO_2$, $SO_2$ and HCHO by other methods at WLG and measurements of $NO_2$, $SO_2$, HCHO and BrO by MAX-DOAS in other remote areas at middle and subtropical latitudes.

**Minor comments**

- I believe that the excessive listing of references at the top of page 4 makes no sense as these references have no connection to the topic of this manuscript.

We have deleted these references in the revised manuscript.

- I'm not so sure about the discussion of straylight on page 8. My guess is, that the straylight correction does not primarily correct for more straylight in the reddish direct sun spectra (cloudy sky spectra should have a similar wavelength distribution as direct sun spectra) but that it corrects for the change in wavelength dependence of Fraunhofer filling in.

There is an important difference between direct sun measurements and scattered light measurements under cloudy skies. The effect of Rayleight scattering on air molecules is opposite:

-for direct sun measurements, Rayleigh scattering leads to a decrease of light at short wavelengths compared to top of the atmosphere sun spectrum.

-for scattered light spectra in cloudy conditions, Rayleigh scattering above the cloud increase the light at short wavelengths.
For measurements in the UV spectral range as used in this study, the difference is substantial.

- It should at least be mentioned that zenith measurements which are used here for the CI were earlier discarded for trace gas retrievals because of suspected direct sun impact
Done. Please see Sect. 5.1 of the revised manuscript.

- Figure 8: Are these realistic stratospheric BrO profiles? This would imply a factor of 4 variability in stratospheric BrO columns – has this been observed in measurements?
Of course, the profiles shown in the figure include extreme scenarios. Nevertheless, variations of more than a factor of 2 are usually found in satellite observations close to and over polar regions, see. e.g. Fig. 9 in Theys et al., 2011. Here it should be noted that the BrO VCDs in the tropics are even smaller and below the minimum of the colour scale used in this figure.
(Theys, N., Van Roozendael, M., Hendrick, F., Yang, X., De Smedt, I., Richter, A., Begoin, M., Errera, Q., Johnston, P. V., Kreher, K., and De Mazière, M.: Global observations of tropospheric BrO columns using GOME-2 satellite data, Atmos. Chem. Phys., 11, 1791–1811, https://doi.org/10.5194/acp-11-1791-2011, 2011.).

- Figure 10: Title of figures not correct
This figure has been moved to the Supplement as Fig. S27. The title was corrected: dAMF => dSCDs

- Figure 11: t does not become clear if all directions are included in the statistics and if so why. My understanding is that only 1 measurements were used in a quantitative way.
Indeed, in Fig. 11 (for old version) all directions were included. We agree with the reviewer that the statistics for $1°$ elevation should be shown. We replaced Fig. 11 accordingly (see Fig. 6 in the revised manuscript). It should be noted that the relative frequencies of the sky conditions stay almost unchanged.

- Conclusions, line 22: Enhanced trace gas mixing ratios – enhanced in comparison to what?
In comparison to the higher elevation angles. We added this information to the text.

- I'm surprised that no reference is made to the paper
Gomez, L., Navarro-Comas, M., Puentedura, O., Gonzalez, Y., Cuevas, E. and Gil-Ojeda, M.: Long-path averaged mixing ratios of $O_3$ and $NO_2$ in the free troposphere from mountain MAX-DOAS, Atmos. Meas. Tech., 7(10), 3373–3386, doi:10.5194/amt-7-3373-2014, 2014. which uses a similar method for a related topic.
We have added this and two other references (Gil-Ojeda et al., 2015; Schreier et al., 2016) in the Introduction and Sect. 6.3 of the revised manuscript, and the results from this study are compared with previous studies, including Gomez et al. (2014).

[revised manuscript text omitted]

Average O$_4$ dSCDs for clear sky conditions

[Figure]

Average O$_4$ dSCDs for broken clouds

[Figure]

**Fig. S27** O$_4$ dSCDs at 1 ° elevation for clear sky **(top)** and broken clouds **(bottom)** spectra (averages of 10 original spectra for 2012 - 2015) with number of scans > 800 and RMS of the O$_4$ fit < 2e-3.

[Figure]

**Fig. S28** Absolute **(left)** and relative **(right)** frequency of the different sky conditions for results of selected trace gases (top: NO$_2$, middle: BrO and HCHO, bottom: SO$_2$). The statistics are based on the number of observations at 1° elevation angle (mean of 10 original spectra from April 2012 to April 2015). In addition to the filter for the removal of high aerosol loads, also the specific RMS filters for the different trace gases are applied (see Supplement Sect. 3.3).

**7 Seasonal means of the dSCDs under broken cloud conditions**

[Figure]

**Figure S29** Seasonal means of the trace gas dSCDs for different elevation angles for broken clouds and low aerosol load. For $NO_2$, $SO_2$, and HCHO the right axes represent the approximate mixing ratios for measurements at $1°$ elevation angle. The blue dotted lines indicate the systematic uncertainties, which can be considered as lower bound of the detection limit.

**8 Estimation of tropospheric BrO mixing ratios from the measured BrO dSCDs**

Fig. S30 shows the dependence of the measured BrO dSCDs for different elevation angles together with simulation results for tropospheric background mixing ratios of 0 ppt or 1 ppt. While for 1 °elevation a strong dependence on the tropopause height is found, such a dependency is not observed for the higher elevation angles. These findings are consistently found for the measurements and simulations (but the measured BrO dSCDs show a rather large scatter). Interestingly, the differences between the simulated BrO dSCDs for 0 and 1 ppt tropospheric background are almost constant. This finding indicates that the increase of the measured BrO dSCDs (with respect to a simulation without a tropospheric BrO background) caused by an increase of the tropospheric BrO background is almost proportional to the tropospheric BrO mixing ratio.

For the estimation of the (upper limit of the) tropospheric BrO background the measurements at 1 °elevation angle are chosen, because they are most sensitive to BrO absorptions in the troposphere (the difference between the simulations for 0 and 1 ppt in Fig. S30 are largest for an elevation angle of 1°). To account for the dependence on the tropopause height the following procedure was applied:

First the measured BrO dSCDs are subtracted from the simulated BrO dSCDs for the same tropopause height. This is done for the simulations for 0 ppt as well as 1 ppt background BrO. From the obtained differences the man values and the standard deviations are calculated. For 0 ppt the mean difference is $0.9*10^{13}$ molec/cm², for 1 ppt it is $2.4*10^{13}$ molec/cm². The standard deviation for both differences is $1.0*10^{13}$ molec/cm².

In the next step the total error (of the difference between simulations and measurements) is calculated as the sum of the standard deviation and the estimate for the systematic error of $0.3*10^{13}$ molec/cm² (see Supplement Sect. 3.3.3).

In the final step the total error ($1.3*10^{13}$ molec/cm²) is compared to the mean values of the differences for 0 ppt and 1 ppt. The total error is found above the mean difference for the simulations for 0 ppt BrO ($0.9*10^{13}$ molec/cm²), and below the mean difference for the simulations for 1 ppt BrO ($2.4*10^{13}$ molec/cm²). Assuming that the (increase of the) measured BrO dSCD depends linearly on the BrO background mixing ratio (see above), an upper limit for the BrO background mixing ratio of 0.23 ppt is obtained. From similar calculations for 6 ° and 16 °elevation, upper limits of 0.34 ppt and 0.60 ppt are found, respectively.

[Figure]

**Fig. S30** Comparison of measured and simulated BrO dSCDs as function of the tropopause height for clear sky conditions. The blue and magenta lines represent simulation results for 1 ppt and 0 ppt BrO in the troposphere, respectively.

---

## Author Response (AR2)

We thank the editor and anonymous referee for further carefully reviewing our manuscript and providing constructive comments and suggestions, which have helped to improve the quality of our manuscript. Below are our point-to-point responses to these comments and suggestions. Also enclosed is the revised version of the manuscript, with the changes marked up with revision track.

Editor and referee comments are in black. Author responses are in blue.

**Editor comments**

One referee had some unsettled issues with regard to interpretations of the measurements and three technical aspects. Please consider addressing them.
We have addressed theses issues carefyully. Please see below.

I also think that the paper would be stronger with more discussion on the results.
We performed chemical box model simulations and correspondingly added a new subsection ″6.4 Simulations of chemical ozone production and OH concentrations″ with more discussions on the results.

Within the current framework, perhaps the author can highlight any novel development of the measurement/retrieval method compared to others and comment on the prospect of using MAX-DOAS technique for background composition measurements (given very low concentrations), remaining challenges and further improvement etc.
The following text was added at the end of the conclusions:
   *Within this study, existing retrieval strategies for MAX-DOAS measurements were adapted and improved for measurements at high altitude stations and in environments with low trace gas abundances. These improvements will be important to similar studies and include the following main aspects:*
   *a) In order to achieve low detection limits, spectra were pre-filtered and averaged to minimise the spectral noise. Spectral interferences between different absorbers were investigated and minimised using measured and synthetic spectra. Maximum wide spectral ranges were used to make best use of the information content.*
   *b) Radiative transfer simulations were performed taking into account the surface topography. While it turned out for this study, that the effects of surface topography were not very important, this might be different for measurements in other mountainous scenarios, especially for measurements on isolated mountains. These effects should be investigated in more detail in future studies.*
   *c) At high mountain sites the elevation angle dependence of stratospheric absorptions can become important, especially for trace gases like BrO which have their concentration maximum close to the tropopause.*
   *An obvious conclusion from our study is that future measurements should be performed with more elevation angles and with instruments having a better signal to noise ratio.*

On the measurement results, can comment on how the new NO2 data shed light on the previous issue of net ozone production or loss at this site.

We calculated the summertime net ozone production, $P_{O3}$, using a chemical box model constrained by different $NO_2$ concentration values, including that drived by MAX-DOAS in this study (a daytime average of 60 ppt $NO_2$ in summer). The results show that there is a little of net ozone loss (-0.8 ppb $d^{-1}$) for the free tropospheric condition (M20) and a little of net ozone production (0.3 ppb $d^{-1}$) for the boundary layer condition (M20-BL). Plesae see Sect. 6.4 for details.

I have a few additional comments on some parts in the added comparison with previous measurements at the site.
Page 11, line 4-5, on statement of positive bias in NO2 measured by CLS/PLC. To my knowledge, there is no known significant bias from photolytic conversion (PLC) of NO2 to NO. It is the commonly used chemical conversion method (using MoO catalyst) that can cause large positive bias by converting not only NO2 but also HNO3, PAN and other oxidized nitrogen. So I think the second reason suggested in the paper is likely a cause.
Agreed. We have skipped over the first reason in the revised manuscript.

Page 11, line 15-17, on HCHO: I think HCHO at the WLG site can be largely from oxidation of isoprene emitted from vegetations, so its source is at the surface, which explains higher concentrations measured previously at the site.
Agreed. We have changed this part to the following text:
*HCHO forms through various oxidation reactions associated with methane and various NMVOCs, most of which (e.g., isoprene) have high spatial variability in abundance due to short lifetimes and thus can influence local HCHO concentrations significantly. Therefore, isoprene and other very active NMVOCs emitted from natural vegetation can accumulate near the surface and result in high levels of HCHO by oxidation. While in situ sampling method as used by Mu et al. (2007) measured HCHO on the ground level within the boundary layer, the MAX-DOAS measured the HCHO mixing ratios at a large scale, partly including the lower free troposphere. Model sensitivity experiment shows that the HCHO mixing ratio is larger in the boundary layer than in the free troposphere over WLG (see Sect. 6.4).*

Minor points:
Page 2, line 34-35, "modify the incomplete sentence "similar to…"
The sentence has been changed to "*Similar to $NO_2$, other reactive gases (e.g., $SO_2$ and HCHO) were also measured by filter or canister sampling method at WLG at irregular time*".

Page 11, line 20, change "had be" to "had been".
Done.

Page 12, line 13, change "approved" to "proved".
Done.

**Reviewer comments**

In their revised version, the authors have moved a lot of material into the supplement making the main article much more accessible for readers. They have also added a

comparison of their measurements with previous observations at the same site and high altitude site observations from other stations reported in the literature. This provides context to the results which has been missing in the original manuscript.

Unfortunately, no attempt was made to add some interpretation or scientific questions to be answered which would have made this a much more valuable study.

There are still three points from my original review which I think the authors need to address:

1) The authors added interesting figures on 2d-RTM calculations but do not at all explain or discuss them in the text. A dedicated section needs to be added to the text as promised in the reply to my review.
The existing text about the 2D effects was extended and put into a new subsection 4.1:

**4.1 Effects of topography**
We investigated the effect of the surface topography in more detail: In addition to the set-up with the 'true' topography (Fig. 3), we also performed simulations with flat surfaces at sea level or 3700m altitude, because these options might be used as alternative scenarios for radiative transfer models without the option to consider the true topography. In all three set-ups the atmospheric properties and the altitude of the detector (3800m) are kept the same. In Fig. 4 the $O_4$ AMFs for the simulations with flat surfaces are plotted versus the $O_4$ AMFs for the true surface topography. While the results for a flat surface at 3700m are very similar to those for the true topography, the results for a flat surface at sea level show systematically higher values. With increasing aerosol load these differences even increase. This finding can become very important for the interpretation of measurements in mountainous environments if no radiative transfer simulations considering the true topography are available. However, this finding needs further investigations, especially if an instrument is operated on a more isolated mountain. In such cases, the agreement between simulations with true topography and flat surface at high altitude might be worse. Further studies should also investigate the effects for trace gases with different altitude profiles.
Besides the simulation of the air mass factors, we also compared the results for other simulated quantities: In Fig. 5 the dependence of the simulated radiance (top) $O_4$ AMF (center) and colour index (bottom) are shown as function of the elevation angle (including negative elevation angles). Again, the results for the true topography and the flat surface at 3700m altitude are very similar, while the results for a flat surface at sea level are systematically different. The results of these comparisons indicate that for our measurements the assumption of a flat surface at the approximate altitude of the surrounding terrain is a very good approximation for the true surface topography. However, for zero and negative elevation angles, the consideration of the true topography might become important. Here it is interesting to note that these differences will probably increase for measurements in other mountainous scenarios, especially or more measurements at isolated mountains. Also these effects should be investigated in more detail in future studies.

2) While it is good to hear that the authors trust their pointing correction to be accurate within 0.5 degrees, this information should be added to the article and some kind of uncertainty estimate needs to be added to the mixing ratios derived, including the effect from pointing uncertainty.

The effect of the pointing uncertainty was investigated by new simulation studies. The results were added to section 4.1 in the supplement.
The uncertainties range between ±20% (for $NO_2$) to <1% for BrO.

3) There was a misunderstanding with respect to my question about the enhanced mixing ratios. The authors now changed the sentence in the conclusions to "For NO2 and HCHO higher dSCDs are found for lower elevation angles than for higher elevation angles, indicating enhanced trace gas mixing ratios in the lower troposphere over WL". First of all, this still does not answer die question of relative to what the mixing ratios are enhanced (a climatology? other altitudes? expectations?).
The sentence has been changed to:
*For $NO_2$ and HCHO higher dSCDs are found for lower elevation angles than for higher elevation angles, indicating higher trace gas concentrations in the lower troposphere compared to the upper troposphere over WLG.*

In addition, the sentence now is in my opinion not strictly correct - lower elevation angles have longer light paths and therefore usually larger dSCDs. Whether or not this indicates larger trace gas mixing ratios at a certain altitude can only be determined by some kind of profile inversion algorithm. In the opposite situation (smaller dSCDs at low angles), the situation is clearer as discussed for BrO.
Many thanks for bringing up this important point! While it is not possible to perform full profile inversions in this study, it is still possible to compare the measured elevation angle dependence of the trace gas dSCDs with the simulation results for different profile shapes. Such a comparison was included for $NO_2$ and HCHO in the revised version of our manuscript (new Fig. S20). While the comparison is done in a rather simplistic way (only long term averages are used), they clearly indicate that for $NO_2$ and HCHO the concentrations in the atmospheric layers close to the instrument have to by systematically higher than in the upper troposphere to achieve agreement with the measurements. These results including the modified profiles for $NO_2$ and HCHO are included in section 4.1 of the supplement.

Using the updated profile shapes leads also to slightly modified relationships between the measured $NO_2$ and HCHO dSCDs at 1 ° and the corresponding mixing ratios:

*$NO_2$: a dSCD of $1\times10^{15}$ molec/cm ² corresponds to a mixing ratio of 23 ppt (constant mixing ratio) or 33 ppt (modified profile)*
*HCHO: a dSCD of $1\times10^{15}$ molec/cm ² corresponds to a mixing ratio of 42 ppt ppt (constant mixing ratio) or 55 ppt (modified profile)*

All corresponding numbers in the manuscript (including the scales in Fig. 8) were updated accordingly.

[revised manuscript text omitted]

*It should be noted that in addition to the original cross section also a modified cross section including the $O_4$ absorption band at 328 nm was tested (see Lampel et al., 2018)). It was found that the BrO and HCHO dSCDs retrieved using the modified $O_4$ cross section were almost identical (deviations < 2%) with the results from the analysis with the original cross section.

**Table S2d** Fit settings for the $O_4$ spectral analyses.

| $O_4$ analysis | |
|---|---|
| Wavelength range (nm) | 352 – 387 |
| DOAS polynomial | degree: 5 |
| Intensity offset | degree: 2 |
| Gaps (nm) | - |
| Ring effect | Original and wavelength-dependent Ring spectrum |
| $O_4$ | 293 K, Thalman and Volkamer (2013) |
| $NO_2$ | 220 K, Vandaele et al. (2002) |
| $O_3$ | 223 K, Io corrected, Bogumil et al. (2003) |

**3.3.1 Determination of the optimum fit range for HCHO and BrO**

For this task both synthetic and measured spectra are used. Different tests are performed to find the best suited fit range for both species. The results are summarized in Table  S3 below. Based on these results, a fit range from 314 to 358 nm was chosen. The individual tests are described in more detail below.

**Table  S3** Best fit ranges based on different test results.

| Test | Optimum lower fit limit for HCHO (nm) | Optimum upper fit limit for HCHO (nm) | Optimum lower fit limit for BrO (nm) | Optimum upper fit limit for BrO (nm) |
|---|---|---|---|---|
| Comparison with input values of synthetic spectra | 313 – 314 | 356 – 360 | 314 – 316 | 356 – 360 |
| Consistency between synthetic and measured spectra | 314 - 316 | 358 – 359 | 314 – 316 | 357 – 358 |
| Fit error (in brackets: results for synthetic spectra) | 312 – 317 | 358 | 312 – 318 (312 – 316) | 357 – 358 (358 – 360) |
| RMS (in brackets: results for synthetic spectra) | 316 – 318 (314 – 318) | 356 – 358 (356 – 360) | 316 – 318 (314 – 318) | 356 – 358 (356 – 360) |
| scatter of results for 1° elevation angle (in brackets: results for synthetic spectra) | 316 (312 – 317) | 358 – 360 (356 – 358) | 315 – 316 (312 – 313) | 358 – 360 (358 – 360) |
| correlation between BrO and HCHO dSCDs for 1° elevation angle (in brackets: results for synthetic spectra) | 312 – 313 (313 – 318) | 358 – 360 (312 – 313) | 312 – 313 (313 – 318) | 358 – 360 (312 – 313) |
| | | | | |
| **Final selection** | **314** | **358** | **314** | **358** |

*Synthetic spectra*

Synthetic spectra were simulated at high spectral resolution for the spectral range 303 – 390 nm using the RTM SCIATRAN. Rotational Raman scattering was included. The simulations were performed for a SZA of 50 ° and a relative azimuth angle (RAA) of 180 °. Surface albedo and altitude were set to 0.07 and 3800m, respectively.

An aerosol layer between 3800 and 4800m with an AOD of 0.1 was assumed. The single scattering albedo and phase function were chosen according to biomass burning aerosols. For the ozone absorption the temperature dependence was taken into account. Information about the chosen trace gas cross sections and assumed atmospheric profiles is given in Table S4.

The Radiance output is convoluted with a Gaussian function with FWHM of 0.6 nm. Random noise with a RMS of 5e-4 is added to the convoluted spectra. 100 spectra with different noise are simulated for each elevation angle.

In addition to the simulated spectra, also air mass factors are derived from the RTM for the following wavelengths: 315, 340, 355 nm. The resulting dSCDs for BrO and HCHO are shown in Fig. S6. The dSCDs are calculated assuming a Fraunhofer reference spectrum measured at 26° elevation angle. The dSCDs derived in this way are compared to the results of the spectral analyses. For this comparison, the analysis results of the 100 spectra for each elevation angle are averaged. For BrO and HCHO the dSCDs calculated for 340 nm are used for the comparison.

**Table  S4** Trace gas cross sections and atmospheric profiles used for the synthetic spectra

| Trace gas | Cross section | Atmospheric profile |
|---|---|---|
| $NO_2$ | NO2_vandaele97_220_vac.txt | Box profile in the lowest 0.5km. VCD: 1e15 molec/cm²; stratospheric profile with maximum at 24 km. VCD: 5.22e15 molec/cm² |
| HCHO | HCHO_Meller_298_vac.DAT | Box profile in the lowest 1km. VCD: 1e15 molec/cm² |
| BrO | bro_wil_228_vac.txt | stratospheric profile with maximum at 20 km. VCD: 3e13 molec/cm² |
| $O_3$ | O3_203K_V3_0.dat
O3_223K_V3_0.dat
O3_243K_V3_0.dat
O3_273K_V3_0.dat

[revised manuscript text omitted]

**4 Input trace gas profiles for the radiative transfer simulations**

**4.1. NO₂, SO₂ and HCHO**

For these trace gases tThe SO₂ vertical concentration profile is are determined assuming a constant mixing ratio throughout the atmosphere. Of course this is a rather strong simplification of the true profiles, which usually have much more complex shapes. But from the measurements, no information about the true atmospheric SO₂ profile can be derived, because the measured SO₂ dSCDs are below the detection limit. Nevertheless, based on this simple assumption it is still possible to estimate the approximate tropospheric trace gas mixing ratios of NO₂, upper limit or the tropospheric SO₂, mixing ratio from the measured SO₂ dSCDs. For NO₂ and HCHO we assumed two profile shapes: first, like for SO₂ a constant mixing ratio throughout the troposphere was assumed. Second, modified profile shapes were chosen (red lines in Fig. S18), which fit best to the measured elevation angle dependence of both trace gases (see Fig. S20). For both trace gases it became obvious that the measured elevation angle sequence can only be (approximately) matched if enhanced trace gas concentrations are present in the layer close to the instrument (between 3700m and 4300m). For HCHO the best match with the measured elevation dependence is even found if enhanced HCHO mixing ratios are only present in these layers. from the corresponding measured trace gas dSCDs. Fortunately, for the estimation of the NO₂ and HCHO mixing ratios, the exact profile assumptions are not critical. The relationships between the measured trace gas dSCDs at 1° elevation and the corresponding mixing ratios in the atmospheric layer close to the instrument are very similar:

**NO₂:** a dSCD of $1\times10^{15}$ molec/cm² corresponds to a mixing ratio of 23 ppt (constant mixing ratio) or 33 ppt (modified profile)

**HCHO:** a dSCD of $1\times10^{15}$ molec/cm² corresponds to a mixing ratio of 42 ppt ppt (constant mixing ratio) or 55 ppt (modified profile)

In Fig. S18 the trace gas concentration profiles used in the RTM simulations are shown. They are calculated for typical background mixing ratios of the trace gases. However, it should be noted that the exact knowledge of the true mixing ratio is not important, because for these weak atmospheric absorbers, the air mass factors are almost independent from the absolute trace gas concentrations.

[Figure]

**Fig. S18** Vertical concentration profiles of $SO_2$, HCHO, and $NO_2$ used in the radiative transfer simulations. For HCHO and $NO_2$ two profiles were used: One profile (blue) assumes a constant mixing ratio in the troposphere, the other (red) shows profiles which fit best to the measured elevation angle dependence (see Fig. S20).  In the figures also the corresponding vertical column densities (VCDs) are given.

In Fig. S19 the trace gas dSCDs of $NO_2$, $SO_2$, and HCHO corresponding to the  input profiles with constant mixing ratios in the troposphere are shown as a function of the elevation angle. The trace gas dSCDs are given for different aerosol loads and for different seasons (left: summer; right: winter). Interestingly, for HCHO and $SO_2$, the dSCDs at $1°$ elevation angle are almost independent from the aerosols load. In contrast, for the $NO_2$ dSCDs at $1°$ elevation angle a larger dependence on the aerosol load is found, probably because of less Rayleigh scattering at these

longer wavelengths. Nevertheless, for simulations with AODs between 0.1 and 0.5, also the $NO_2$ dSCDs at $1°$ elevation angle very similar.

[Figure]

5  **Fig. S19** Trace gas dSCDs simulated for the profiles with constant mixing ratios shown in Fig. 6 for summer **(left)** and winter **(right)**. The individual lines represent results for different aerosol loads.

Figure S20 compares the average elevation angle dependence of the measured $NO_2$ and HCHO dSCDs (see also Fig. 8) with simulated dAMFs for different profile assumptions. From this comparison it becomes obvious that the simulations for constant mixing ratios do not fit the measured elevation dependences. The discrepancies are especially large for HCHO indicating that most of the HCHO resides in atmospheric layers close to the instrument. Here it should be noted that no perfect agreement should be expected, because of the rather simplistic comparison (using averaged results from all seasons). Nevertheless, as mentioned above, the conversion of the measured $NO_2$ and HCHO dSCDs into trace gas mixing ratios in the atmospheric layers close to the instrument is only weakly affected by the profile assumptions.

We also investigated the effect of the uncertainty of the elevation calibration (+/-0.5 °) on the simulated dAMFs by performing simulations with modified elevation angles. Changes of the elevation angle by +/- 0.5 ° have a very small effect on the dAMFs for high elevation angels (<2%) and are therefore negligible. For 1 ° elevation angles the effects can become larger and are given below for the different trace gases and profile assumptions:

$SO_2$, constant mixing ratio:     <5%

$NO_2$, constant mixing ratio:     <5%

$NO_2$, modified profile:            <20%

HCHO, constant mixing ratio: <5%

HCHO, modified profile:         <15%

BrO                             <1%

[Figure]

**Fig. 20** Average elevation angle dependence of HCHO (left) and $NO_2$ (right) for measurements at clear sky and low aerosol load. Black lines: measured average dSCDs (left axes); red: simulated dAMFs assuming constant mixing ratios in the troposphere; blue: simulated dAMFs for modified trace gas profiles (see Fig. S18).

**4.2 BrO**

[revised manuscript text omitted]

**Fig.** S26 BrO dSCDs calculated for the profiles shown in Fig. S24C with 1 ppt BrO in the troposphere for different tropopause heights (**left:** winter; **right:** summer).

**6 Cloud and aerosol filters**

[Figure]

**Fig. S26 S27** Simulated O$_4$ dAMFs for different aerosol loads for winter **(left)** and summer **(right)**. Constant aerosol extinction was assumed between 2600 and 5600m.

Average O$_4$ dSCDs for clear sky conditions

[Figure]

Average O$_4$ dSCDs for broken clouds

[Figure]

**Fig.**  **S28** O$_4$ dSCDs at 1 °elevation for clear sky **(top)** and broken clouds **(bottom)** spectra (averages of 10 original spectra for 2012 - 2015) with number of scans > 800 and RMS of the O$_4$ fit < 2e-3.

[Figure]

Fig.  S29 Absolute **(left)** and relative **(right)** frequency of the different sky conditions for results of selected trace gases (top: NO$_2$, middle: BrO and HCHO, bottom: SO$_2$). The statistics are based on the number of observations at 1 ° elevation angle (mean of 10 original spectra from April 2012 to April 2015). In addition to the filter for the removal of high aerosol loads, also the specific RMS filters for the different trace gases are applied (see Supplement Sect. 3.3).

**7 Seasonal means of the dSCDs under broken cloud conditions**

[Figure]

**Figure**  **S30** Seasonal means of the trace gas dSCDs for different elevation angles for broken clouds and low aerosol load. For $NO_2$, $SO_2$, and HCHO the right axes represent the approximate mixing ratios for measurements at $1°$ elevation angle. The blue dotted lines indicate the systematic uncertainties, which can be considered as lower bound of the detection limit.

**8 Estimation of tropospheric BrO mixing ratios from the measured BrO dSCDs**

Fig. S31 shows the dependence of the measured BrO dSCDs for different elevation angles together with simulation results for tropospheric background mixing ratios of 0 ppt or 1 ppt. While for 1 °elevation a strong dependence on the tropopause height is found, such a dependency is not observed for the higher elevation angles. These findings are

5 consistently found for the measurements and simulations (but the measured BrO dSCDs show a rather large scatter). Interestingly, the differences between the simulated BrO dSCDs for 0 and 1 ppt tropospheric background are almost constant. This finding indicates that the increase of the measured BrO dSCDs (with respect to a simulation without a tropospheric BrO background) caused by an increase of the tropospheric BrO background is almost proportional to the tropospheric BrO mixing ratio.

10 For the estimation of the (upper limit of the) tropospheric BrO background the measurements at 1 °elevation angle are chosen, because they are most sensitive to BrO absorptions in the troposphere (the difference between the simulations for 0 and 1 ppt in Fig. S31 are largest for an elevation angle of 1 °). To account for the dependence on the tropopause height the following procedure was applied:

First the measured BrO dSCDs are subtracted from the simulated BrO dSCDs for the same tropopause height. This

15 is done for the simulations for 0 ppt as well as 1 ppt background BrO. From the obtained differences the man values and the standard deviations are calculated. For 0 ppt the mean difference is $0.9*10^{13}$ molec/cm $^2$, for 1 ppt it is $2.4*10^{13}$ molec/cm $^2$. The standard deviation for both differences is $1.0*10^{13}$ molec/cm $^2$.

In the next step the total error (of the difference between simulations and measurements) is calculated as the sum of the standard deviation and the estimate for the systematic error of $0.3*10^{13}$ molec/cm $^2$ (see Supplement Sect. 3.3.3).

20 In the final step the total error ($1.3*10^{13}$ molec/cm $^2$) is compared to the mean values of the differences for 0 ppt and 1 ppt. The total error is found above the mean difference for the simulations for 0 ppt BrO ($0.9*10^{13}$ molec/cm $^2$), and below the mean difference for the simulations for 1 ppt BrO ($2.4*10^{13}$ molec/cm $^2$). Assuming that the (increase of the) measured BrO dSCD depends linearly on the BrO background mixing ratio (see above), an upper limit for the BrO background mixing ratio of 0.23 ppt is obtained. From similar calculations for 6 ° and 16 ° elevation, upper limits of

25 0.34 ppt and 0.60 ppt are found, respectively.

[Figure]

**Fig. S31** Comparison of measured and simulated BrO dSCDs as function of the tropopause height for clear sky conditions. The blue and magenta lines represent simulation results for 1 ppt and 0 ppt BrO in the troposphere, respectively.